# S6K1 phosphorylates Cdk1 and MSH6 to regulate DNA repair

Adi Amar-Schwartz[1†], Vered Ben Hur[1†], Amina Jbara[1†], Yuval Cohen[1], Georgina D Barnabas[2], Eliran Arbib[3], Zahava Siegfried[1], Bayan Mashahreh[1], Fouad Hassouna[1], Asaf Shilo[1], Mohammad Abu-Odeh[3], Michael Berger[3], Reuven Wiener[1], Rami Aqeilan[3], Tamar Geiger[2], Rotem Karni[1]*

[1]Department of Biochemistry and Molecular Biology, the Institute for Medical Research Israel-Canada, Hebrew University-Hadassah Medical School, Jerusalem, Israel; [2]Department of Human Molecular Genetics and Biochemistry, Sackler Faculty of Medicine, Tel Aviv University, Tel Aviv, Israel; [3]Lautenberg Center for Immunology and Cancer Research, Department of Immunology and Cancer Research, the Institute for Medical Research Israel-Canada, Hebrew University-Hadassah Medical School, Jerusalem, Israel

*For correspondence: rotemka@ekmd.huji.ac.il

†These authors contributed equally to this work

Competing interest: The authors declare that no competing interests exist.

**Abstract** The mTORC1 substrate, S6 Kinase 1 (S6K1), is involved in the regulation of cell growth, ribosome biogenesis, glucose homeostasis, and adipogenesis. Accumulating evidence has suggested a role for mTORC1 signaling in the DNA damage response. This is mostly based on the findings that mTORC1 inhibitors sensitized cells to DNA damage. However, a direct role of the mTORC1-S6K1 signaling pathway in DNA repair and the mechanism by which this signaling pathway regulates DNA repair is unknown. In this study, we discovered a novel role for S6K1 in regulating DNA repair through the coordinated regulation of the cell cycle, homologous recombination (HR) DNA repair (HRR) and mismatch DNA repair (MMR) mechanisms. Here, we show that S6K1 orchestrates DNA repair by phosphorylation of Cdk1 at serine 39, causing G2/M cell cycle arrest enabling homologous recombination and by phosphorylation of MSH6 at serine 309, enhancing MMR. Moreover, breast cancer cells harboring *RPS6KB1* gene amplification show increased resistance to several DNA damaging agents and S6K1 expression is associated with poor survival of breast cancer patients treated with chemotherapy. Our findings reveal an unexpected function of S6K1 in the DNA repair pathway, serving as a tumorigenic barrier by safeguarding genomic stability.

## Editor's evaluation

This study provides evidence for a novel function of S6K1 in DNA repair. The findings explain how amplification of RPS6KB1 gene, which encodes the S6K1 protein, in breast cancers is associated with a poor outcome after chemotherapy. The conclusions of the paper are well-supported by the experimental results. The proteomics data leading to the discovery of Cdk1 and Msh6 as targets of S6K1 are novel and important for the cancer research field.

## Introduction

The mTOR complex 1 (mTORC1) is a central regulator of cellular metabolism and biosynthesis, and is subjected to complex regulation by growth factors, nutrients, oxygen availability and cellular stress (*Raught et al., 2004*). When nutrients are available, mTORC1 phosphorylates its two main substrates, S6K1 (S6 Kinase 1) and 4E-BP1/2 (eukaryotic initiation factor 4E (eIF4E)-binding proteins 1 and 2), which are at the helm of cell growth, survival, translation, and proliferation (*Mamane et al., 2006*).

**eLife digest** Damage to the DNA in our cells can cause harmful changes that, if unchecked, can lead to the development of cancer. To help prevent this, cellular mechanisms are in place to repair defects in the DNA. A particular process, known as the mTORC1-S6K1 pathway is suspected to be important for repair because when this pathway is blocked, cells become more sensitive to DNA damage.

It is still unknown how the various proteins involved in the mTORC1-S6K1 pathway contribute to repairing DNA. One of these proteins, S6K1, is an enzyme involved in coordinating cell growth and survival. The tumor cells in some forms of breast cancer produce more of this protein than normal, suggesting that S6K1 benefits these cells' survival. However, it is unclear exactly how the enzyme does this.

Amar-Schwartz, Ben-Hur, Jbara et al. studied the role of S6K1 using genetically manipulated mouse cells and human cancer cells. These experiments showed that the protein interacts with two other proteins involved in DNA repair and activates them, regulating two different repair mechanisms and protecting cells against damage.

These results might explain why some breast cancer tumors are resistant to radiotherapy and chemotherapy treatments, which aim to kill tumor cells by damaging their DNA. If this is the case, these findings could help clinicians choose more effective treatment options for people with cancers that produce additional S6K1. In the future, drugs that block the activity of the enzyme could make cancer cells more susceptible to chemotherapy.

Recent studies have pointed to a distinction between the mTORC1-S6K1 and the mTORC1-4EBP axes, arguing that the mTORC1-S6K1 axis regulates cell size and glucose metabolism with minor effects on translation and proliferation, while the mTORC1-4EBP axis appears to have a greater impact on translation, proliferation, and tumor formation (*Dowling et al., 2010*; *Whittaker et al., 2010*; *Hsieh et al., 2010*; *von Manteuffel et al., 1997*; *Ben-Hur et al., 2013*; *Ruvinsky et al., 2005*; *Shima et al., 1998*; *Ruvinsky and Meyuhas, 2006*; *Pende et al., 2004*).

S6K1 is known for its ability to coordinate nutritional status and control cell growth (*Ruvinsky et al., 2005*; *Shima et al., 1998*; *Magnuson et al., 2012*) but its role in tumorigenesis remains elusive, even controversial. *RPS6KB1*, the gene encoding for S6K1, is located in chromosomal region 17q23 which is amplified in 8.8% of primary breast cancers (*Bärlund et al., 2000*). However, this amplification covers over 4 MB and contains nearly 50 genes (*Sinclair et al., 2003*) including *Tbx2*, *PRKCA*, *Tlk2*, *TUBD1*, and *PPM1D*, that are known for their correlation to breast cancer and/or contribution to oncogenic signaling (*Sinclair et al., 2003*; *Olson et al., 2011*; *Abrahams et al., 2008*; *Lu et al., 2005*; *Ali et al., 2012*; *Groth et al., 2003*; *Xu et al., 2008*; *Monni et al., 2001*), making estimation of the contribution of S6K1 amplification to the disease difficult. Our previous work had identified a cancer promoting alternative splicing switch that modulates between S6K1 long and short variants (*Ben-Hur et al., 2013*; *Karni et al., 2007*). While the short isoform of S6K1, which lacks kinase activity (termed here Iso-2), acts as an oncogenic driver and activates the mTORC1-4EBP axis, the long S6K1 variant (termed here S6K1) was shown to harbor tumor suppressive activity (*Ben-Hur et al., 2013*). This activity can only partially be explained by the known regulatory negative feedback loop of the Akt-mTORC1 signaling pathway mediated by S6K1 phosphorylation of the Insulin Receptor Substrate Protein 1 (IRS1) (*Efeyan and Sabatini, 2010*).

The mTORC1-S6K1 arm is best known for its role in nutrient-dependent signaling pathways underlying cell growth, and less for its role in DNA damage response (DDR). However, several reports have demonstrated that genotoxic stress regulates mTORC1 (*Zhou et al., 2017*). Sensing of DNA damage by mTORC1 is mediated through several pathways; Ras/MEK/ERK (*Lai et al., 2010*; *Braunstein et al., 2009*), ATM/LKB1/AMPK (*Alexander et al., 2010*) or other yet undiscovered mechanisms. mTOR inhibition has also been suggested to sensitize cancer cells to ionizing radiation (IR) or to interstrand cross-linkers (*Albert et al., 2006*; *Cao et al., 2006*; *Shen et al., 2013*; *Bae-Jump et al., 2009*). These reports suggest involvement of mTORC1 in DNA repair, though no formal proof has been established. Recently, it has been proposed that in response to double-strand breaks (DSBs), the mTORC1-S6K1 pathway transcriptionally regulates a key component of the Fanconi Anemia (FA) pathway ultimately

reducing phosphorylated histone 2 A variant X (γ-H2AX) (*Shen et al., 2013*), a well-known surrogate marker of DSBs (*Bonner et al., 2008*). It was also reported that the insulin−IGF-1−PI3K−AKT pathway is implicated in DDR (*Matsuoka et al., 2007*) and S6K1, as well as Akt and other components of this pathway, were found to be phosphorylated on their ATM and ATR consensus sites in response to IR (*Matsuoka et al., 2007*). Finally, S6K1 itself was implicated in DDR. Cells from S6K1/S6K2 double knockout mice show increased DNA damage, without any stimulus, as detected by increased γ-H2AX (*Zhou et al., 2017*).

The gene encoding for S6K1, *RBS6KB1* is amplified in 8–10% of breast cancer tumors (*Bärlund et al., 2000*). However, we previously reported that the long kinase active isoform of S6K1 acts as a tumor suppressor in certain conditions when it is overexpressed while the short kinase-dead isoform promotes oncogenesis (*Ben-Hur et al., 2013*). These intriguing results are typical of genes that, on one hand, control an essential cellular process while, on the other hand, are an obstacle for certain cancer cells or reduce their fitness. BRCA1 and BRCA2 are good examples of genes whose loss promotes cancer development but also make the cancer cell vulnerable to DNA damage. Thus, we sought to examine if and how S6K1 is involved in the DNA damage response and in DNA damage repair. Using mass spectrometry analysis of proteins associated with S6K1, we identified interactions between S6K1 and components of the DNA repair machinery including; Proliferating Cell Nuclear Antigen (PCNA), MutS homolog 2 (MSH2), MutS homolog 6 (MSH6), Cyclin Dependent Kinase 1(Cdk1, also known as CDC2) and others. We demonstrate that the latter two (MSH6 and Cdk1) are direct substrates of S6K1. We show that S6K1 functions as a DNA damage checkpoint protein for both HRR and MMR mechanisms: Inducing G2/M cell cycle arrest by Cdk1 phosphorylation thus allowing HR repair and enhancing MMR by phosphorylating and activating MSH6. These findings suggest that the mTORC1-S6K1 axis may serve as an important path in modulating DDR to ensure genomic integrity during proliferation and upon exposure to increased DNA hazards.

## Results

### S6K1 protects cells from various DNA damaging agents

Several reports have demonstrated that genotoxic stress regulates mTORC1 (*Bärlund et al., 2000*; *Braunstein et al., 2009*) and imply involvement of mTORC1 in resistance to DNA damage (*Zhou et al., 2017*; *Albert et al., 2006*; *Cao et al., 2006*; *Shen et al., 2013*). We set out to examine whether up-regulation of S6K1 expression protects cells from DNA damaging agents and confers resistance to chemotherapeutic drugs. To this end, we overexpressed either the full length active kinase S6K1 (S6K1) or the S6K1 short isoform (Iso-2) in S6K1/S6 K2 double knockout (S6K$^{-/-}$) MEFs (*Figure 1a and b*). We compared viability and survival of these cells upon induction of DNA damage. Expression of S6K1 (but not Iso-2) rescued S6K$^{-/-}$ MEF cells from DNA damage induced cell death following treatment with either doxorubicin (Dox), γ irradiation, UV irradiation or neocarzinostatin (NCS) (*Figure 1c–j*). Overexpression of S6K1 in MCF-10A cells protected, while its knockdown sensitized the cells to cell death following treatment with either Dox, γ irradiation, UV-irradiation, NCS or cisplatin (CDDP) (*Figure 1—figure supplement 1a-i*). Moreover, inhibition of mTORC1 by rapamycin, inhibited the protective effect of S6K1 following Dox treatment (*Figure 1k*, *Figure 1—figure supplement 1k*). These results suggest that S6K1 protects cells from different types of DNA damage in an mTORC1-dependent manner. Protection from DNA damage by S6K1 was accompanied by reduced histone H2AX serine 139 phosphorylation (γ-H2AX), a surrogate marker for DSBs (*Bärlund et al., 2000*) while sensitization to DNA damaging agents following S6K1 knockdown was associated with higher levels of γ-H2AX (*Figure 1l and m*, *Figure 1—figure supplement 1j*). Under these experimental conditions we also observed reduced levels of cleaved caspase-3 and cleaved poly (ADP-ribose) polymerase 1) PARP-1(in cells overexpressing S6K1 (*Figure 1l–m*, *Figure 1—figure supplement 1j*). These results suggest that S6K1 confers resistance to various types of DNA damage.

### S6K1 expression is a marker for poor prognosis in breast cancer patients treated with chemotherapy and protects cells harboring *RPS6KB1* amplification from chemotherapeutic damage

In order to determine if these findings reflect what has been reported in clinical data, we analyzed the correlation between S6K1 expression and probability of survival in two breast cancer patient

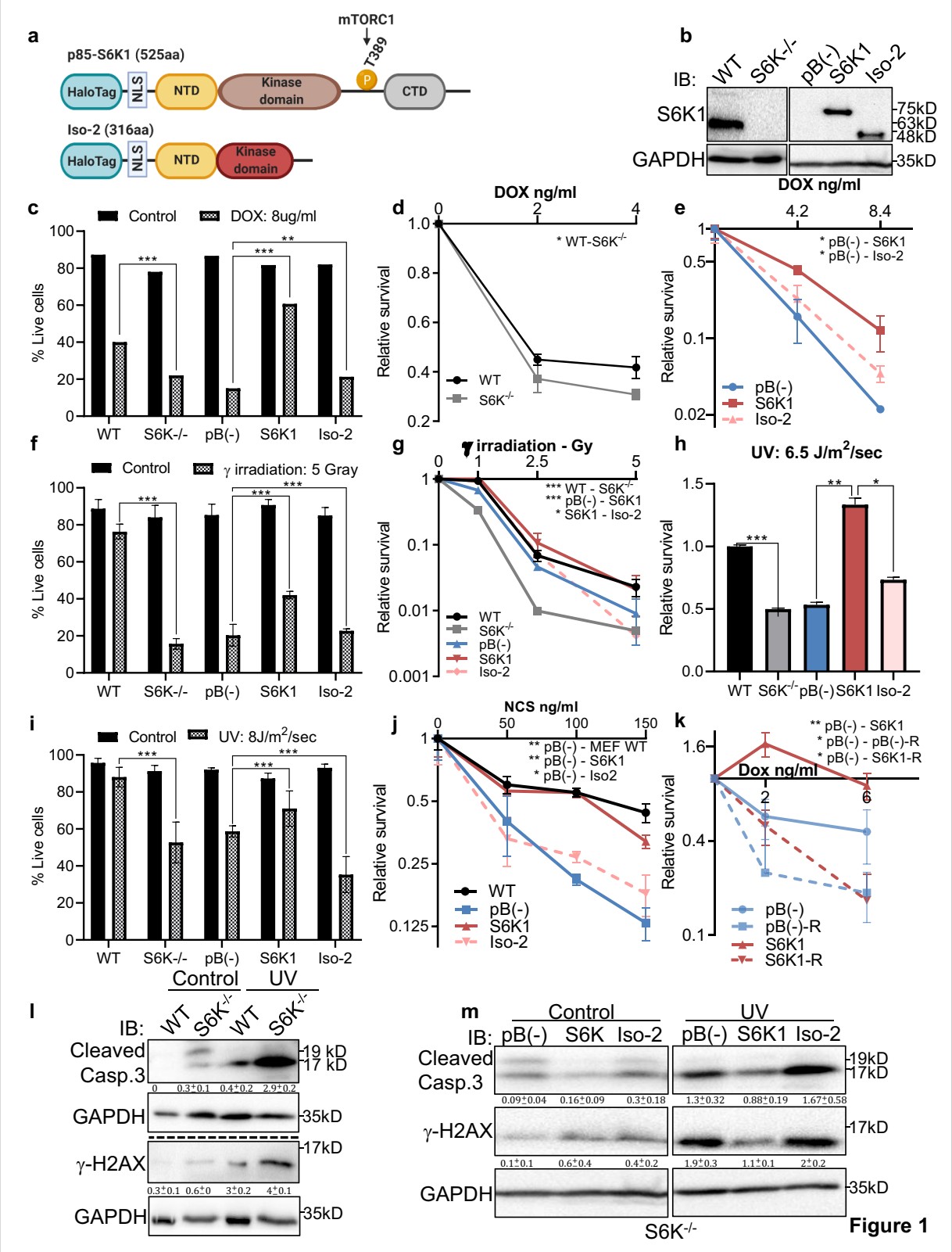

**Figure 1.** S6K1 protects cells from DNA damaging agents. (**a**) Long (S6K1) and short (Iso-2) S6K1 isoform structure and domain organization with addition of Halo-tag (light blue ovals). Isoform Iso-2 lacks 6 out of 12 conserved regions of the kinase domain, a mTOR phosphorylation site at threonine 389 (T389), as well as the C terminal autoinhibitory domain (CTD). NLS (nuclear localization sequence), NTD (N terminal domain). (**b**) Western blot showing the expression of S6K1 in MEF wild type (WT), S6K-/- cells and S6K-/- cells transduced with retroviruses encoding either empty vector pBABE

*Figure 1 continued on next page*

*Figure 1 continued*

(pB(-)), S6K1 or Iso-2. (**c,f,i**) Trypan blue exclusion assay of MEF WT, S6K-/- and S6K-/- cells described in (**b**) treated with doxorubicin 8 µg/ml (DOX) for 24 hr (**c**) or 24 hr after either γ irradiation 5 Gy (**f**) or UVC 8 J/m$^2$/s (**i**). Data represents means ± SD of six replicates. (**d,e,g,h,j**) Clonogenic survival assay of MEF WT, S6K-/- cells (**d,g,h,j**), and S6K-/- cells transduced with retroviruses encoding either empty vector pBABE (pB(-)), S6K1 or Iso-2 (**e,g,h,j**) treated with various concentrations of Dox (**d,e**), γ irradiation 1–5 Gy (**g**), UVC 6.5 J/m$^2$/s (**h**) or various concentrations of neocarzinostatin (NCS) (**j**). Colonies were fixed and quantified after 14 days. Data represents means ± SD of at least three biological replicates. (**k**) Clonogenic survival assay of S6K-/- cells expressing either empty vector pBABE (pB(-))or S6K1. Cells were treated with various concentrations of doxorubicin (DOX) with and without 50 nM rapamycin (**R**). Data represents means ± SD of three biological triplicates. (**l,m**) Western blot analysis of MEF WT and S6K-/- cells (**l**) or cells described in (**b**) (**m**) 24 hours after irradiation with UVC 6.5 J/m$^2$/s. Cleaved caspase 3 and γ–H2AX signals were quantified relative to GAPDH signal. * p<0.05, ** p<0.01, *** p<0.001. p values were calculated using Student's t-test (two-tailed). Data represents means ± SD of three biological triplicates.

The online version of this article includes the following source data and figure supplement(s) for figure 1:

**Source data 1.** Western blots of *Figure 1*.

**Figure supplement 1.** S6K1 protects MCF-10A cells from DNA damaging agents.

**Figure supplement 1—source data 1.** Western blots of *Figure 1—figure supplement 1*.

**Figure supplement 2.** RPS6KB1 expression is a poor prognosis marker for breast cancer patients and cell lines treated with chemotherapy.

**Figure supplement 2—source data 1.** Western bots of *Figure 1—figure supplement 2*.

**Figure supplement 3.** S6K1 does not affect ATM levels or phosphorylation state.

**Figure supplement 3—source data 1.** Western blots of *Figure 1—figure supplement 3*.

populations: Patients treated with chemotherapy and patients that were never treated with chemotherapy. In breast cancer patients that were never exposed to chemotherapy, S6K1 expression had no correlation with probability of survival, while in the treated patient population S6K1 expression was a significant poor prognosis marker (*Figure 1—figure supplement 2a, b*). These results led us to hypothesize that when S6K1 is up-regulated it confers drug resistance and reduces treatment efficacy resulting in reduced survival. Given that the gene encoding S6K1, *RPS6KB1*, is amplified in about 8–10% of breast cancer patients (*Bärlund et al., 2000*), we decided to examine the sensitivity of breast cancer cell lines, with and without *RPS6KB1* amplification, to chemotherapeutic agents. Two breast cancer cell lines harboring *RPS6KB1* amplification (MCF7, BT474) were compared to two breast cancer cell lines with low S6K1 expression (HCC70, SKBR3) for sensitivity to chemotherapeutic agents. Cells harboring an *RPS6KB1* amplification were significantly more resistant to CDDP and doxorubicin treatments than cells with no amplification, as determined by trypan blue exclusion and Caspase 3 and PARP-1 cleavage, both markers of chemotherapy-induced apoptosis (*Figure 1—figure supplement 2c-e*).

## S6K1 does not affect ATM levels or phosphorylation state

A previous study showed that S6K1/S6K2 signaling suppresses the levels of ATM by induction of miRNA RNA-18a and microRNA-421 and that mTOR/S6K inhibition or S6K1/2 knockout elevates ATM protein levels (*Shen and Houghton, 2013*). Thus, we examined ATM and p-ATM levels both in MEFs (WT, S6K-/-, and overexpression of S6K1 isoforms) and in MCF-10A cells (S6K1 knockdown by shRNAs, overexpression of S6K1 isoforms) before/after treatment with either gamma irradiation or UV (*Figure 1—figure supplement 3*). We did not observe any change in ATM levels, due to a change in S6K1 levels, in any of the different cells or conditions we examined. We did detect, as expected, elevated ATM phosphorylation following DNA damage induction (*Figure 1—figure supplement 3*). These results differ from what was reported by *Shen and Houghton, 2013* regarding S6K1/2 knockout cells and suggest that, in our system, S6K1 effects on γ-H2AX phosphorylation and DNA damage-induced cell death are not mediated by ATM signaling.

## S6K1 interacts with DNA damage response components and phosphorylates Cdk1 and MSH6

To explore the possible mechanisms by which S6K1 protects cells from DNA damage, we sought to examine if S6K1 interacts directly with proteins involved in the DNA damage response and/or DNA repair. HEK293 cells were transfected with plasmids containing different S6K1 isoforms tethered to a halo-tag (HT) (*Figure 1a*, Materials and methods). Protein pulldown assays and two independent mass-spectrometry analyses were performed on these cells. As phosphorylation usually causes

conformational changes of the substrate due to steric repulsion of the negatively charged phosphate groups, we hypothesized that the S6K1 kinase inactive variant (*Ben-Hur et al., 2013*), short isoform Iso-2, may have a stronger interaction with all putative substrates. We therefore included this isoform in the pulldown assay. Iso-2 has a truncated kinase domain and lacks the folded conformation created by the interaction with the auto-inhibitory C-terminal domain (CTD) (*Magnuson et al., 2012*), allowing availability of the N-terminal region for protein-protein interactions (*Figure 1a*). Combined proteomic analyses (described in detail in Materials and methods), identified approximately 400 putative interactors, of which 164 were identified by both mass-spectrometry pull-down assays (*Figure 2—figure supplement 1a*, *Supplementary file 1*). As predicted, many more interactions were detected with the short isoform (Iso-2) than with S6K1 (*Figure 2a*, *Supplementary file 1*). Analysis of the S6K1 and Iso-2 bound proteins revealed proteins that participate in, and are responsible for, DNA repair, vesicular mediated transport and mitochondrial transport or mitochondrial electron transport (*Supplementary file 1*). These proteins included five known S6K1 interacting proteins from the literature (compiled in the BioGrid tool - https://thebiogrid.org/); S6K1, mTOR, UBL4A, XPO1, and HSP90AA1 (*Figure 2—figure supplement 1c*, *Supplementary file 2*). A previous study that identified S6K1 and S6K2 interactors identified 21 high-scoring S6K1 interactions, of which two (S6K1, HSP90AA1) were also found in our study (*Pavan et al., 2016*; *Figure 2—figure supplement 1b*, *Supplementary file 2*). We chose to focus on the set of proteins in the DNA damage response/repair module (*Figure 2a*). We validated the binding of several of these proteins to S6K1 by HT-pulldown followed by immunoblot (*Figure 2b–d*) or reciprocal immunoprecipitation (*Figure 2f*, *Figure 2—figure supplement 1e-i*). Indeed, Cdk1 and proteins of the MMR machinery, such as MutSα complex (MSH2 and MSH6) and PCNA, were found to bind to either S6K1, Iso-2 or both and did not bind to HT control (*Figure 2b–f*). This was observed in both HeLa cells, for proteins MSH2, PCNA, and MCM7 (*Figure 2d*) and HEK293 cells, for proteins MSH2 and PCNA (*Figure 2b and c*). Immunoblot of PCNA immunoprecipitated lysates showed interaction of PCNA with both S6K1 and Iso-2 in HEK293 cells (*Figure 2f*) and in HeLa cells (*Figure 2—figure supplement 1f, g*). Immunoblot of MSH6 immunoprecipitated lysates showed interaction of MSH6 with Iso-2 in HeLa cells (*Figure 2—figure supplement 1h, i*). Co-immunoprecipitation of Cdk1 only showed interaction with HT-Iso-2 (*Figure 2e*). It should be noted that the interaction with Iso-2 is stronger due to lack of steric repulsion caused by S6K1 phosphorylation. Taken together, these findings point towards a possible involvement of S6K1 in DDR signaling and/or DNA repair (*Matsuoka et al., 2007*; *Polo and Jackson, 2011*).

Most of the DDR proteins are modified by phosphorylation upon DNA damage which facilitates their recruitment to damage sites and modulates their DNA repair activity (*Polo and Jackson, 2011*). Using the PhosphoNET browser (http://www.phosphonet.ca/default.aspx) we searched for putative S6K1 phosphorylation motifs RX(RXXS) (*Ruvinsky and Meyuhas, 2006*) in the validated S6K1 interacting proteins. Cdk1 and MSH6, but not PCNA, have multiple putative S6K1 phosphorylation sites (*Supplementary file 3*). To examine if Cdk1 and MSH6 are direct S6K1 substrates, we performed an in vitro kinase assay using full-length Cdk1 and truncated MSH6 recombinant proteins. The truncated MSH6 recombinant protein, comprising of amino acids 1–718, contained the first three domains; the N-terminal disordered domain (NTR), mismatch binding domain and connector domain (*Edelbrock et al., 2013*). These regions have many predicted S6K1 phosphorylation sites (*Supplementary file 3*). The MSH6 NTR was also reported to regulate MMR downstream signaling of DNA damage mediators (*Edelbrock et al., 2013*; *Gassman et al., 2011*). We performed an in vitro kinase assay using radioactive $^{32}$P, incubating recombinant S6K1 alone or in combination with either Cdk1, MSH6 (Materials and methods) or ribosomal S6 protein (rpS6) as a positive control. In addition, we also performed this experiment, without radioactive $^{32}$P, using an antibody that recognizes the phosphorylation motif (RXXS) of the AGC kinase family to detect S6K1 specific phosphorylation. As shown by both radiolabeling and immunoblotting, S6K1 phosphorylated Cdk1 and MSH6 (*Figure 3a and b*). To eliminate the possibility of the presence of a contaminating protein, which may be acting as a kinase or substrate in our assay, we performed the in vitro kinase assay using recombinant Uba5 as a negative control. Uba5 does not contain any RXXS phosphorylation motifs but does have 26 serine and 19 threonine residues. In our assay S6K1 did not phosphorylate Uba5 (*Figure 2—figure supplement 2a, b*). Co-transfection of MSH6 with S6K1 into HEK293 cells followed by MSH6 pull-down revealed enhanced phosphorylation of MSH6 on the RXXS motif, as shown by immunoblotting using the phospho-specific RXXS antibody (*Figure 3c and d*). Furthermore, co-transfection of Cdk1 with S6K1 into HEK293 cells followed

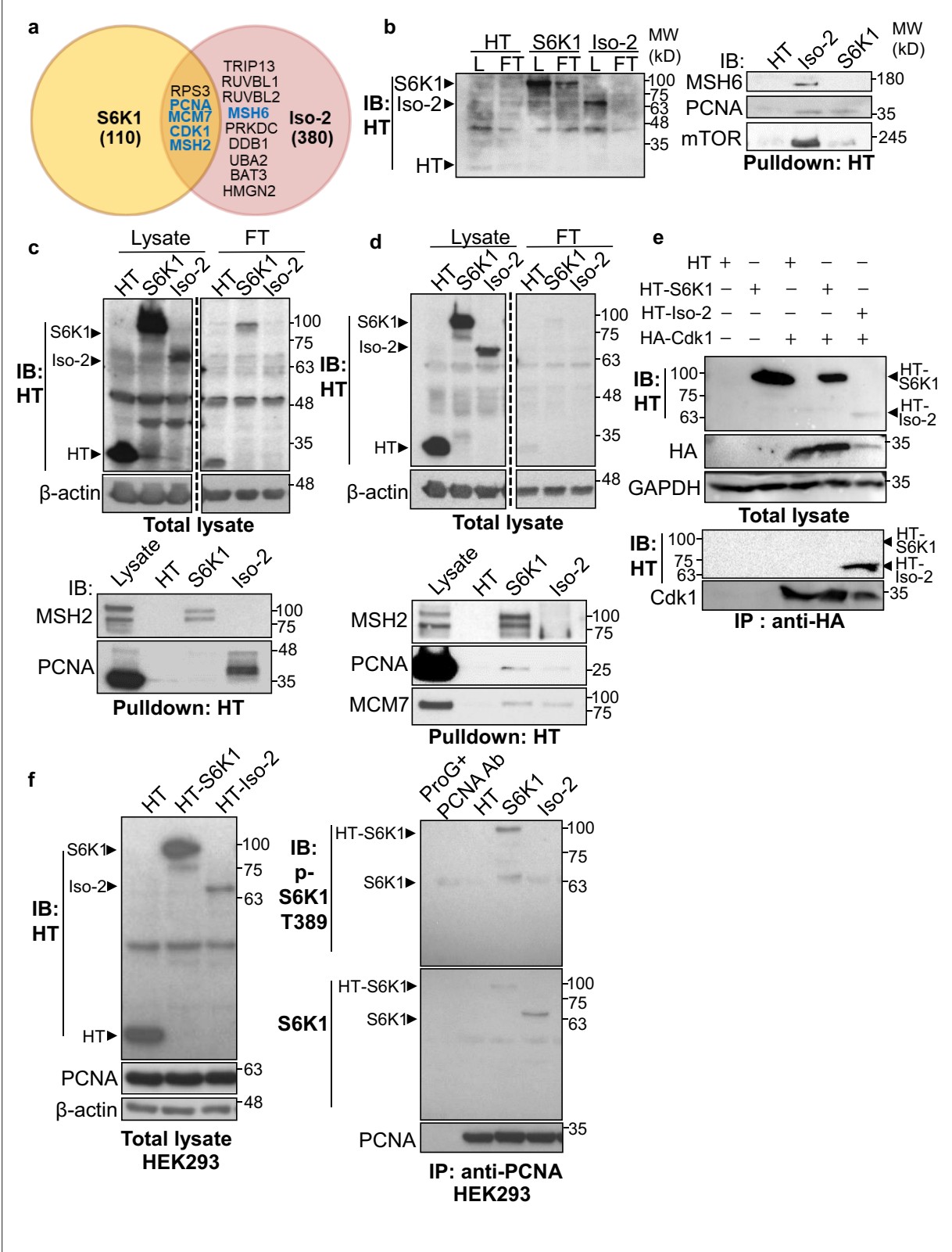

**Figure 2.** S6K1 and Iso-2 bind to DNA damage/repair proteins. (**a**) Venn diagram representing DNA damage response/repair proteins that bound specifically to HT-S6K1 isoforms in transfected HEK293 cells. Proteins marked in blue were validated by reciprocal immunoprecipitations. (**b,c**) Western blot of total lysate (**L**) and flow through (FT) of Halo-tag pulldown from HEK293 cells transfected with either Halo-tag expressing control vector (HT), HT-S6K1 (S6K1) or HT-Iso-2 (Iso-2) (b, left panel and c, upper panel). Western blot of Halo-tag pulldown from the same cells (b, right panel c, lower panel).

*Figure 2 continued on next page*

*Figure 2 continued*

(**d**) Western blot of total lysate (**L**) and flow through (FT) of Halo-tag pulldown from HeLa cells transfected with either Halo-tag expressing control vector (HT), HT-S6K1 (S6K1) or HT-Iso-2 (Iso-2) (upper panel). Western blot of Halo-tag pulldown from the same cells transfected with either Halo-tag expressing control vector (HT), S6K1 or Iso-2 (lower panel). (**e**) Western blot of total lysate from HEK293 cells co-transfected with either Halo-tag expressing control vector (HT), HT-S6K1 (S6K1) or HT-Iso-2 (Iso-2) and HA-Cdk1 (upper panel). Western blot of immunoprecipitation of lysates of these cells with anti-HA antibody (lower panel). (**f**). Western blot analysis of HEK293 cells transfected as described in b (left panel). Western blot of immunoprecipitation of lysates of these cells with anti-PCNA antibody (right panel).

The online version of this article includes the following source data and figure supplement(s) for figure 2:

**Source data 1.** Western blots of *Figure 2*.

**Figure supplement 1.** S6K1 isoforms interact with proteins involved in DNA repair.

**Figure supplement 1—source data 1.** Western blots of *Figure 2—figure supplement 1*.

**Figure supplement 2.** S6K1 does not phosphorylate unrelated protein Uba5.

by HA-Cdk1 pull-down revealed enhanced phosphorylation of Cdk1 on the RXXS motif, as shown by immunoblotting using the phospho-specific RXXS antibody (*Figure 3e and f*). It should be noted that phosphorylation levels are higher after treatment with UV (*Figure 3e and f*). These results suggest that Cdk1 and MSH6 are direct substrates of S6K1. Using mass-spectrometry on the recombinant proteins after an in vitro kinase assay, we detected Cdk1 phosphorylation on serine 39 and MSH6 phosphorylation on multiple serine and threonine residues (*Supplementary file 3*).

## S6K1 enhances nuclear localization of MSH6 and reduces γ-H2AX foci formation in the nucleus

Previous reports have shown the importance of post translational modifications in MutSα cytoplasmic/nuclear shuttling *Gassman et al., 2011*; *Christmann et al., 2002*. The DDR proteins, which we identified as substrates of S6K1 (*Figure 3*), are mainly localized in the nucleus but can shuttle between the cytoplasm and the nucleus *Hayes et al., 2009*. We therefore examined the subcellular localization of S6K1 substrates upon induction of DNA damage. MCF-10A cells overexpressing S6K1, but not cells overexpressing Iso-2, showed enhanced nuclear localization of MSH6 and MSH2 after treatment with UV irradiation (*Figure 4a and b*). Although the trend was the same for MSH2 and MSH6, only changes in MSH6 subcellular localization were significant (*Figure 4b*). We did not detect any change in total MSH2 and MSH6 levels under these conditions in total cell extracts (*Figure 4—figure supplement 1a*). In line with these results, cells with S6K1 knockdown showed reduced nuclear localization of MSH6 following UV irradiation (*Figure 4c and d*, *Figure 4—figure supplement 1b*). These findings support a role for S6K1 in MMR regulation, by phosphorylation and nuclear translocation of MSH6.

S6K1 proteins exist as both long (p85) and short (p70) variants due to alternative translational start sites, created by an N–terminal extension of 23 amino acids possessing a nuclear localization signal (NLS) (*Grove et al., 1991*; *Fenton and Gout, 2011*). Yet, S6K1 subcellular localization is mainly cytoplasmic with a few reports showing nucleo-cytoplasmic shuttling following adipogenic stimuli depending on mitogen activation or mTORC1 inhibition (*Fenton and Gout, 2011*; *Rosner and Hengstschläger, 2011*; *Rosner et al., 2012*). Subcellular fractionation of MCF-10A cells stably expressing S6K1 showed cytoplasmic localization of S6K1, and no visible nuclear localization of S6K1, both with and without DNA damage (*Figure 4—figure supplement 1c*). In order to better visualize S6K1 subcellular localization following DNA damage, U2OS cells transfected with control vector (HT), S6K1 (HT-S6K1) or Iso-2 (HT-Iso-2) were stained with either HaloTag TMR-Direct or R110-Direct fluorescent ligands, following UV irradiation. Nuclear localization of S6K1 was enhanced following UV irradiation, while Halotag control vector or HT-Iso-2 subcellular localization did not change (*Figure 4e*, *Figure 4—figure supplement 1d*). Co-staining of these cells for γ-H2AX and S6K1 after UV induction showed inverse correlation between S6K1 expression and γ-H2AX foci formation in the nucleus (*Figure 4f*, *Figure 4—figure supplement 1e*). These results are in agreement with our previous observations; decreased levels of γ-H2AX in S6K[/-] MEF cells overexpressing S6K1 after DNA damage (*Figure 1l and m*) and in MCF-10A cells overexpressing S6K1 after DNA damage (*Figure 1—figure supplement 1j*). These results suggest that S6K1 nuclear localization may enhance DNA repair which, in turn, reduces γ-H2AX.

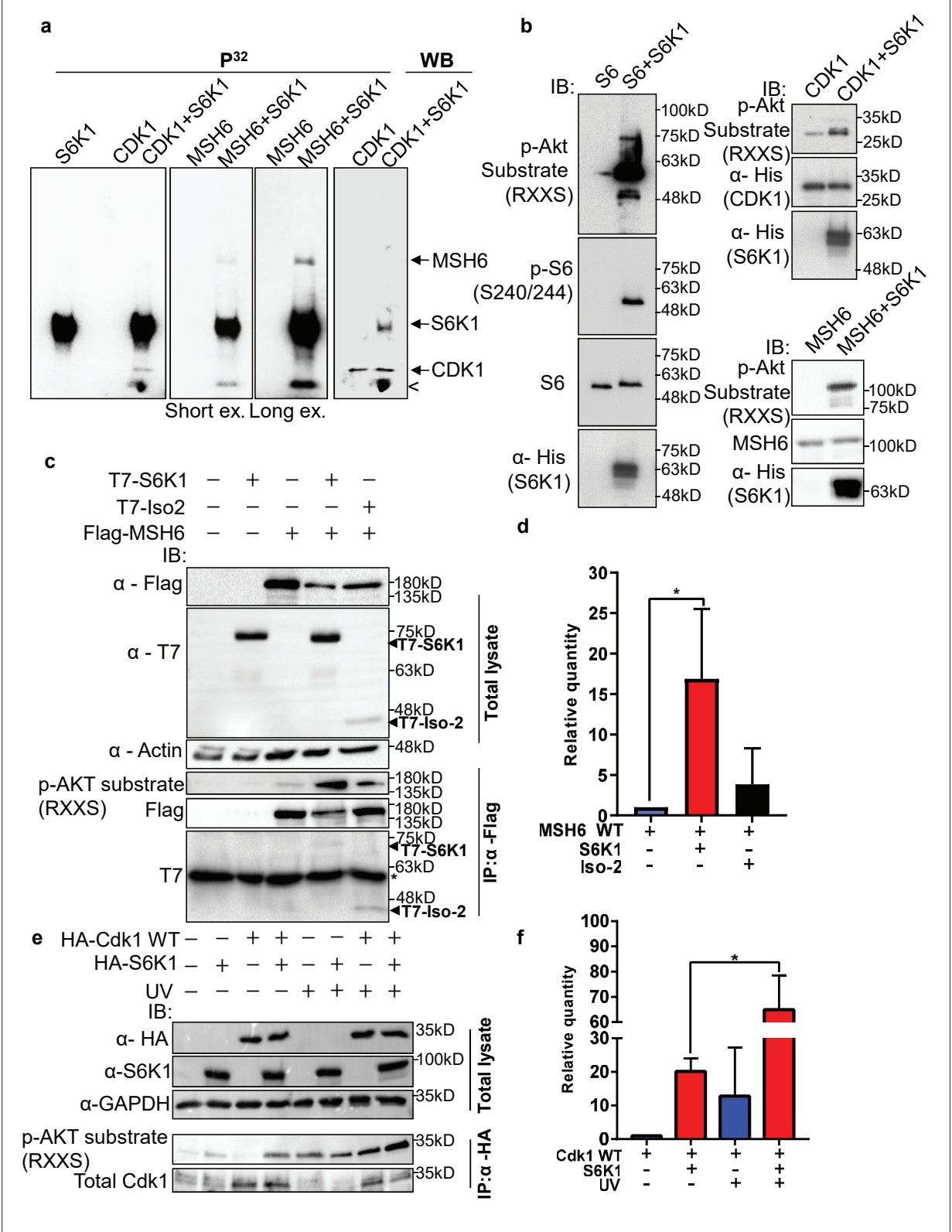

**Figure 3.** S6K1 phosphorylates CDK1 and MSH6 in vitro. (**a**) In vitro kinase assay using γ-$^{32}$P-ATP and recombinant His-S6K1 [T412E] with either recombinant His-Cdk1 (CDK1), or recombinant His-MSH6 (MSH6, 1-718aa fragment as described in Materials and methods) in the indicated combinations. First three panels show reaction products resolved by SDS-PAGE and visualized (the second and third panels are short and long exposures of the same membrane). The fourth panel shows the same membrane as the first panel probed with anti-Cdk1 antibody and detected by

*Figure 3 continued on next page*

*Figure 3 continued*

enhanced chemiluminescence. Nonspecific radioactive band is marked by <. (**b**) Non-radioactive in vitro kinase assay using recombinant His-S6K1 [T412E] with either recombinant GST-S6 (**S6**), His-Cdk1(Cdk1) or His-MSH6 (MSH6). Reaction products were resolved by SDS-PAGE and analyzed by Western Blot using an antibody against the S6K1 phosphorylation motif (p-Akt substrate, RXXS), or antibodies to detect the recombinant proteins. (**c,d**) Western blot analysis of HEK293 cells co-transfected with FLAG-MSH6 and S6K1 isoforms immunoprecipitated with anti-Flag beads. Phosphorylation of MSH6 was detected with p-Akt substrate antibody. Quantification of MSH6 phosphorylation is shown in the graph (**d**). (**e,f**) Western blot analysis of HEK293 cells co-transfected with HA-Cdk1 and S6K1 with and without UVC 30 J/m$^2$ /s (UV) damage isoforms immunoprecipitated with anti-HA beads. Phosphorylation of Cdk1 was detected with p-Akt substrate antibody. Quantification of Cdk1 phosphorylation is shown in the graph (**f**). The experiment was repeated three times (biological replicate). ** $p < 0.01$ p values were calculated using Student's t-test (two-tailed).

The online version of this article includes the following source data for figure 3:

**Source data 1.** Western blots of *Figure 3*.

## S6K1 augments G2/M cell cycle arrest after DNA damage

DNA damaging agents trigger G1/S, intra-S and G2/M cell-cycle checkpoints and cell cycle arrest, which is thought to increase the time available for DNA repair before replication or mitosis ensues (*Jackson and Bartek, 2009*). The Cdk1-cyclin B complex is the main target molecule of the G2/M cell cycle checkpoint (*Smits and Medema, 2001*). In light of the fact that we found that S6K1 binds to and phosphorylates Cdk1 on serine 39 (*Figures 2 and 3*, *Supplementary file 4*) we wanted to determine if this interaction/phosphorylation could induce DNA damage-mediated G2/M arrest (*Yamane and Kinsella, 2005*). In both MCF-10A and S6K$^{-/-}$ MEF cells re-expressing S6K1, an increase in the G2/M fraction was observed following γ irradiation treatment, compared to cells expressing either empty vector or Iso-2 (*Figure 5a–d*, *Figure 5—figure supplement 1a, b*). To better distinguish between the S/G2/M phases, we co-stained the cells with a mitosis-specific marker (p-H3 serine 10), late S phase marker (cyclin A) and G2 phase marker (cyclin B). The use of the cyclin A/B and p-H3 markers allowed us to distinguish between G2 and M fractions. It seems that S6K1/S6K2 are not essential for the G2/M checkpoint as S6K$^{-/-}$ MEFs are not completely defective in their G2/M arrest. However, S6K1 augments the G2 phase checkpoint to enhance cell cycle arrest upon DNA damage (*Figure 5a–d*). These results strongly suggest that S6K1 participates in and affects genotoxic stress induced G2 arrest. To further examine the functional contribution of Cdk1 serine 39 phosphorylation by S6K1, we mutated this residue to alanine (39 A). S6K$^{-/-}$ MEFs over-expressing S6K1 transfected with Cdk1 39 A showed reduced G2/M arrest following γ irradiation, as compared to cells transfected with WT Cdk1 (*Figure 5e and f*). These results suggest that phosphorylation of Cdk1 on serine 39 is required for the induction of G2/M cell cycle arrest following induction of DNA damage by IR.

## S6K1 enhances homologous recombination, probably through Cdk1 phosphorylation

As the preferred repair pathway of double strand breaks in G2/M phase is homologous recombination (HR) (*Kadyk and Hartwell, 1992*; *Johnson and Jasin, 2000*; *Ciccia and Elledge, 2010*), we next sought to test whether S6K1 enhances HR using a system based on the combination of expression of the I-SceI endonuclease and a reporter cassette (*Aparicio et al., 2014*) (*Figure 6a*). U2OS cells expressing a DR-GFP reporter cassette containing mutant copies of GFP and the ISceI recognition sequence (here termed U2OS-DR-GFP cells) (*Pierce et al., 1999*; *Shahar et al., 2012*) were transfected with the ISceI endonuclease. ISceI expression induces DSBs, while HR of the two mutant GFP copies generates a functional GFP with an abolished ISceI site (*Shahar et al., 2012*). U2OS-DR-GFP cells were co-transfected with either HT-S6K1 or HT-Iso2 (or T7-tag S6K1/Iso-2) expression vector and the ISceI endonuclease. We observed increased GFP levels in the S6K1 expressing U2OS-DR-GFP cells as compared to control cells or Iso-2 expressing cells (*Figure 6b*) with no effects on cell cycle (*Figure 6—figure supplement 1a-d*). The fact that Iso-2 failed to induce HR suggests that DNA repair induction is dependent on S6K1 kinase activity. We next knocked out S6K1 in the U2OS-DR-GFP cells by CRISPR/Cas9. Knockout of S6K1 reduced HR as compared to CRISPR control cells (*Figure 6c*). As a positive control we knocked down Rad51, an important component of HR, and as expected, these cells showed reduced HR (*Figure 6d*). In order to examine if phosphorylation of Cdk1 on serine 39 is crucial for HR, we transfected U2OS-DR-GFP cells with either WT Cdk1 or Cdk1 39 A. Cells transfected with Cdk1 39 A showed reduced HR, suggesting that this phosphorylation site is also important for proper HR (*Figure 6e*) with no change in cell cycle (*Figure 6—figure supplement 1e-h*). Furthermore,

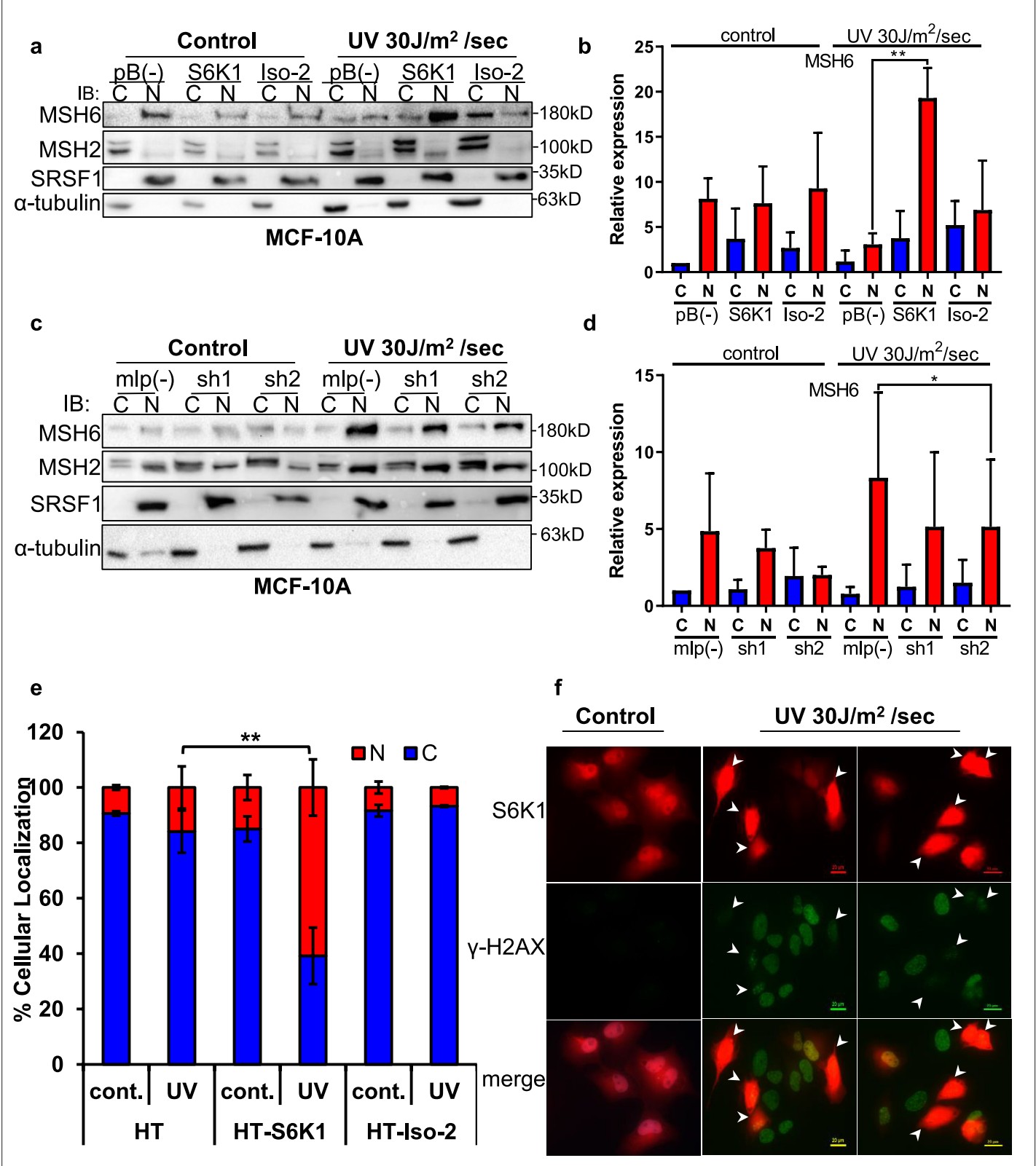

**Figure 4.** S6K1 affects nuclear localization of DNA damage proteins. (**a,b**) MCF-10A cells transduced with either empty vector pBABE (pB(-)), S6K1 or Iso-2 were either untreated (control) or treated with UVC 30 J/m² /sec (UV). Two hours after treatment cells were harvested, fractionated and subjected to western blot analysis. β-tubulin and SRSF1 served as a cytosol (**C**) and nuclear (**N**) markers, respectively. Quantification of the blot is shown in b. Data represents means ± SD of three experiments. (**c,d**) MCF-10A cells transduced with either empty vector (mlp(-)) or S6K1-specific shRNAs (sh1,sh2) were

*Figure 4 continued on next page*

*Figure 4 continued*

either untreated (control) or treated with UVC 30 J/m$^2$ /s (UV). Two hours after treatment cells were harvested, fractionated and subjected to western blot analysis. β-tubulin and SRSF1 served as a cytosol (**C**) and nuclear (**N**) markers, respectively. Quantification of the blot is shown in d. Data represents means ± SD of three biological experiments. (**e**) U2OS cells transfected with either HT, HT-S6K1 (S6K1) or HT-Iso-2 (Iso-2) were either untreated (cont.) or exposed to UVC 30 J/m2 /sec (UV). After 2 hr, cells were fixed and stained with anti-Halo-tag fluorescence TMR Direct ligand and DAPI. Cells were visually scored using a fluorescent microscope for either nuclear staining (**N**) or cytoplasmic staining (**C**). (n=600 cells scored per transfection; 300 for control and 300 for UV treatment). p values were calculated using Student's t-test (two-tailed) from three biological replicate. (**f**) U2OS cells were transfected with HT-S6K1 and exposed to UVC 30 J/m2/s and fixed two hours later. Representative photograph of cells stained with anti-Halo-tag fluorescence TMR Direct ligand (red) and anti-γ-H2Ax (green). Arrows point to S6K1 nuclear localization. Images were taken by Nikon-TL (×20). * p<0.05, ** p<0.01. p Values were calculated using Student's t-test (two-tailed).

The online version of this article includes the following source data and figure supplement(s) for figure 4:

**Source data 1.** Western blots of *Figure 4*.

**Figure supplement 1.** Total lysates of fractionation experiments.

**Figure supplement 1—source data 1.** Western blots of *Figure 4—figure supplement 1*.

when we introduced S6K1 to these cells there were no change in the HR effect (*Figure 6f*) with no change in cell cycle (*Figure 6—figure supplement 1i-l*). Taken together, these results suggest that S6K1-mediated phosphorylation of Cdk1 on serine 39 is required to induce G2/M cell cycle arrest (*Figure 5*) and enable efficient HRR (*Figure 6a–f*).

## S6K1 enhances mismatch DNA repair through MSH6 phosphorylation

Our finding that S6K1 binds to MutSα and PCNA, cofactors that function with the MMR complex (*Jackson and Bartek, 2009*; *Moldovan et al., 2007*; *Figure 2b–h*, *Figure 2—figure supplement 1c-f*), raised the possibility that S6K1 involvement in DNA repair is not restricted only to DSB repair but could also affect MMR. Furthermore, S6K1 phosphorylated MSH6 both in vitro and in cells (*Figure 3*) suggesting a direct regulation of this process. Accumulating evidence suggests that the MMR machinery also contributes to cell cycle arrest induced by DNA damage, as MutSα-deficient cells are defective in cell cycle arrest in response to multiple types of DNA damaging agents (*Yan et al., 2001*; *Stojic et al., 2004*). In order to determine if S6K1 is required for MMR, we employed an assay to measure MMR. In this assay, MMR activity is measured using a plasmid encoding heteroduplex enhanced green fluorescent protein (EGFP) that is expressed only if the G:T mismatch is correctly repaired (*Zhou et al., 2009*). We compared MMR activity of WT MEFs to S6K$^{-/-}$ MEFs. S6K$^{-/-}$ MEFs showed 10-fold lower level of MMR compared to WT cells (*Figure 6g*). To determine if introduction of S6K1 can restore MMR levels in S6K$^{-/-}$ MEFs, we transduced S6K$^{-/-}$ MEFs with either empty vector, S6K1 or Iso-2. S6K$^{-/-}$ MEF cells overexpressing S6K1 showed higher MMR activity compared to cells expressing either empty vector or Iso-2 (*Figure 6h*). Taken together, these results suggest that S6K1 promotes DNA repair, not only through HRR, but also through MMR. To examine if MSH6 phosphorylation by S6K1 can directly regulate MMR, we mutated S6K1 phosphorylation site serine 309 of MSH6 (*Supplementary file 4*) to either alanine (309 A) or aspartate (309D). While overexpression of WT MSH6 did not enhance MMR significantly, expression of the 309 A mutation inhibited MSH6 MMR activity and expression of the 309D mutation, which mimics the phosphorylated state, enhanced MSH6 MMR activity significantly (*Figure 6i*). To our knowledge this is the first demonstration that MSH6 phosphorylation on this site enhances its MMR activity. These results suggest that S6K1 directly enhances MMR activity by MSH6 phosphorylation on serine 309.

## S6K1 phosphorylation of MSH6 309 affects its cellular localization

In order to further investigate the role of MSH6 phosphorylation by S6K1 at position 309, we transfected S6K$^{-/-}$ MEF cells with either Flag-MSH6 WT, Flag-MSH6 309 A or Flag-MSH6-309D mutants. We measure the localization of these constructs following UV irrigation. Nuclear localization of MSH6 309D was enhanced while MSH6 309 A was reduced following UV irradiation (*Figure 7a and b*). These results suggest that MSH6 phosphorylation on serine 309 is crucial for it's localization. Moreover, these results suggest that MSH6 phosphorylation on serine 309 enhances its MMR activity (*Figure 6i*) due to its enhanced nuclear localization (*Figure 7a and b*).

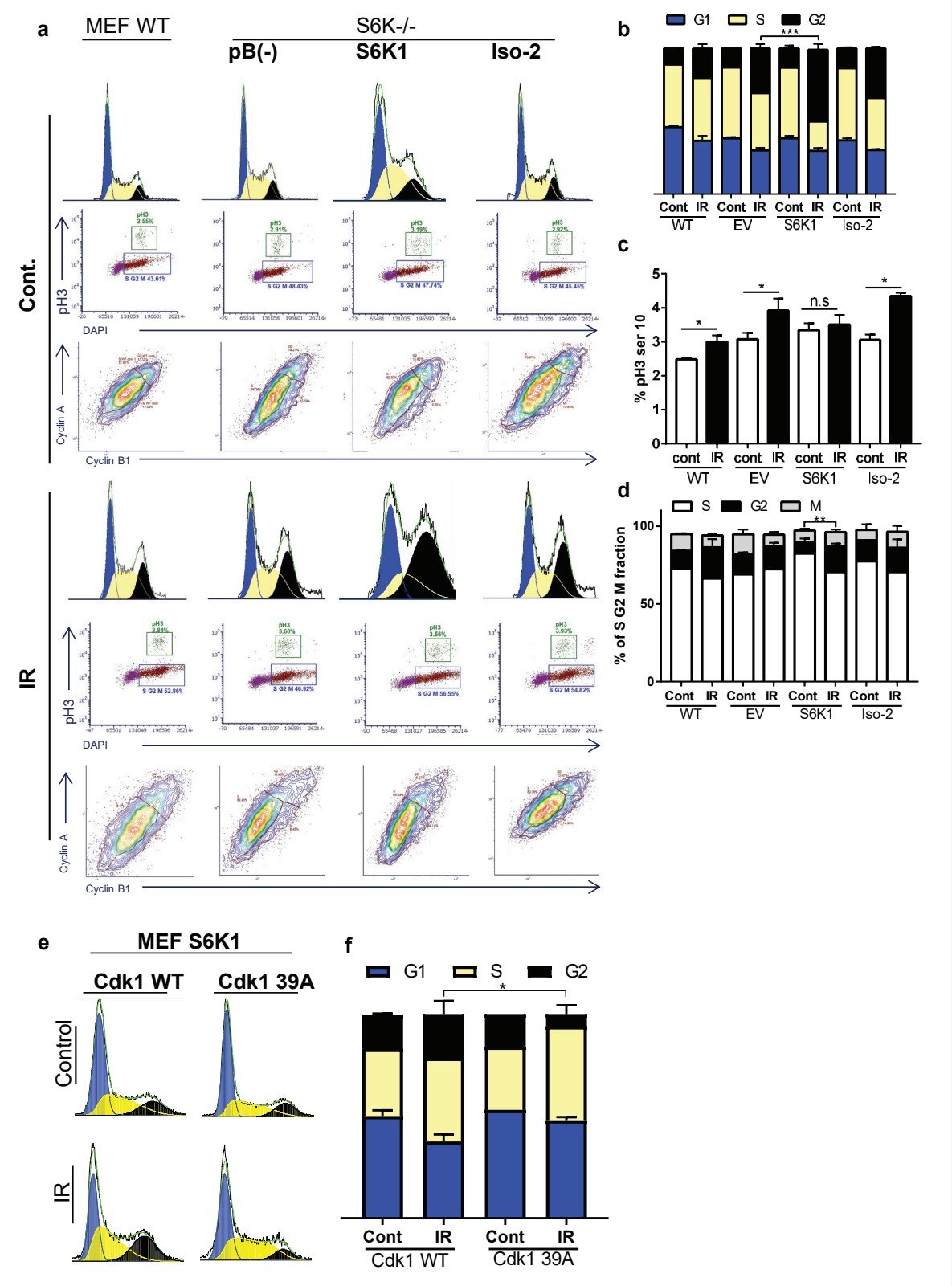

**Figure 5.** S6K1 augments G2/M cell cycle arrest upon DNA damage. (**a–d**) MEF WT or S6K-/- cell populations transduced with retroviruses encoding either empty vector pBABE (pB(-)), S6K1, or Iso-2 were irradiated with ionizing radiation 2.5 Gy (IR) (**a**). Cells were stained with DAPI, anti-phospho-histone H3 Alexa488, anti-Cyclin B1 Alexa647 and anti-Cyclin A PE561. G1, S, and G2 separation displayed according to DAPI staining (**b**). Dual analysis of pH3 and DAPI distinguishes M phase (**c**). Plot was gated on S/G2/M. Dual analysis of cyclin A and cyclin B1 distinguishes between S, G2, M

*Figure 5 continued on next page*

*Figure 5 continued*

(**d**). (**e,f**) Cell cycle FACS analysis of S6K-/- cells expressing retroviruses encoding S6K1 transfected with HA-tagged WT CDK1 or HA-tagged mutant CDK1 39 A. Thirty-six hr after transfection cells were treated with IR 5 Gy. After 24 hr, the cells were subjected to FACS analysis (**e**). The percentage of cells in G1, S and G2/M are depicted in the graph(**f**). * p<0.05, p values were calculated using Student's t-test (two-tailed) from three biological replicate.

The online version of this article includes the following figure supplement(s) for figure 5:

**Figure supplement 1.** Cell cycle analysis of MCF-10A cells.

## Discussion

The mTORC1 signaling pathway regulates many cellular processes involved in cell growth, proliferation and survival which can promote tumorigenicity (*Mamane et al., 2006*; *Hsieh et al., 2010*; *Ben-Hur et al., 2013*; *Dancey, 2010*). Here, we identified a novel direct role for the mTORC1 substrate, S6K1, in regulating HRR and MMR through its interactions with two new putative substrates, Cdk1 and MSH6 (*Figure 7c*). This novel S6K1 function might explain how tumors of breast cancer patients with *RPS6KB1* gene amplification may present as radiation and chemotherapy resistant.

### Enhanced S6K1 expression protects cells from DNA damaging agents and correlates with poor survival of breast cancer patients treated with chemotherapy

Our findings that S6K1 enhances cellular resistance to a variety of DNA insults, involving different types of damage response pathways and multiple repair mechanisms (*Ciccia and Elledge, 2010*), may have clinical relevance. Specifically, S6K1 knockdown or knockout sensitizes cells to treatments with UV radiation, NCS, IR, Dox, and cisplatin. Expression of S6K1 increased cell resistance to these genotoxic treatments and this was accompanied by reduced DNA damage, as detected by lower levels of γ-H2AX, and reduced apoptosis, as detected by reduced cleaved caspase-3 and cleaved PARP1 (*Figure 1* and *Figure 1—figure supplement 1*). The prognostic value of this is supported by the finding that S6K1 expression level correlates with worse survival of breast cancer patients who received chemotherapy but not patients that were never treated with chemotherapy (*Figure 1—figure supplement 2a,b*). Evidence for a causal relationship between S6K1 amplification and enhanced chemotherapy resistance is shown by the fact that breast cancer cell lines harboring *RP6SKB1* amplification, and overexpression of S6K1, show enhanced resistance to chemotherapeutic agents compared to breast cancer cell lines with no amplification and low S6K1 expression (*Figure 1—figure supplement 2c-e*).

S6K2, the paralog enzyme, which is highly similar to S6K1 in its activities, might be involved in some of the interactions and activities we describe here for S6K1. S6K2 has been shown to share some common substrates with S6K1 (*Pavan et al., 2016*). Even though we cannot exclude the possibility that S6K2 mediates some of the described activities, the fact that S6K1 expression could rescue the phenotypes observed in S6K1 and S6K2 double knockout cells (*Figures 1 and 4–6* and *Figure 1—figure supplement 1*) and that S6K1 knockdown/knockout can mimic the phenotypes of the double knockout cells, suggests that S6K1, and not S6K2, is the prominent regulator of the DDR.

### S6K1 interacts with DDR components and phosphorylates MSH6 and Cdk1

Except for several cytoplasmic substrates involved in translational regulation and signaling pathways (*Laplante and Sabatini, 2012*), very little is known about the cellular substrates of S6K1. Only recently, the first nuclear substrate of S6K1, H2B, was identified, connecting S6K1 with chromatin modulation in adipogenesis (*Yi et al., 2016*). Using a combination of pull-down assays and mass spectrometry, we have revealed novel and unexpected interactions between S6K1 and a number of DDR components (*Figure 2*, *Figure 2—figure supplement 1* and *Supplementary file 1*). A previous study by Pavan et al. identified several S6K1 and S6K2 interactors. However, none of the high-scoring S6K1 interactors reported were HRR or MMR components (*Pavan et al., 2016*). Among the low-scoring S6K1 interactors identified were PCNA and MSH6, but not CDK1 or MSH2. Three DDR related proteins identified in that study (PARP1, XRCC6, and XRCC5) bound only S6K2 but not S6K1 in an IP validation experiment (*Pavan et al., 2016*). We compared our S6K1 interactors with those identified in both Pavan et al. and a proteomic database and found surprisingly little overlap (*Figure 2—figure supplement*

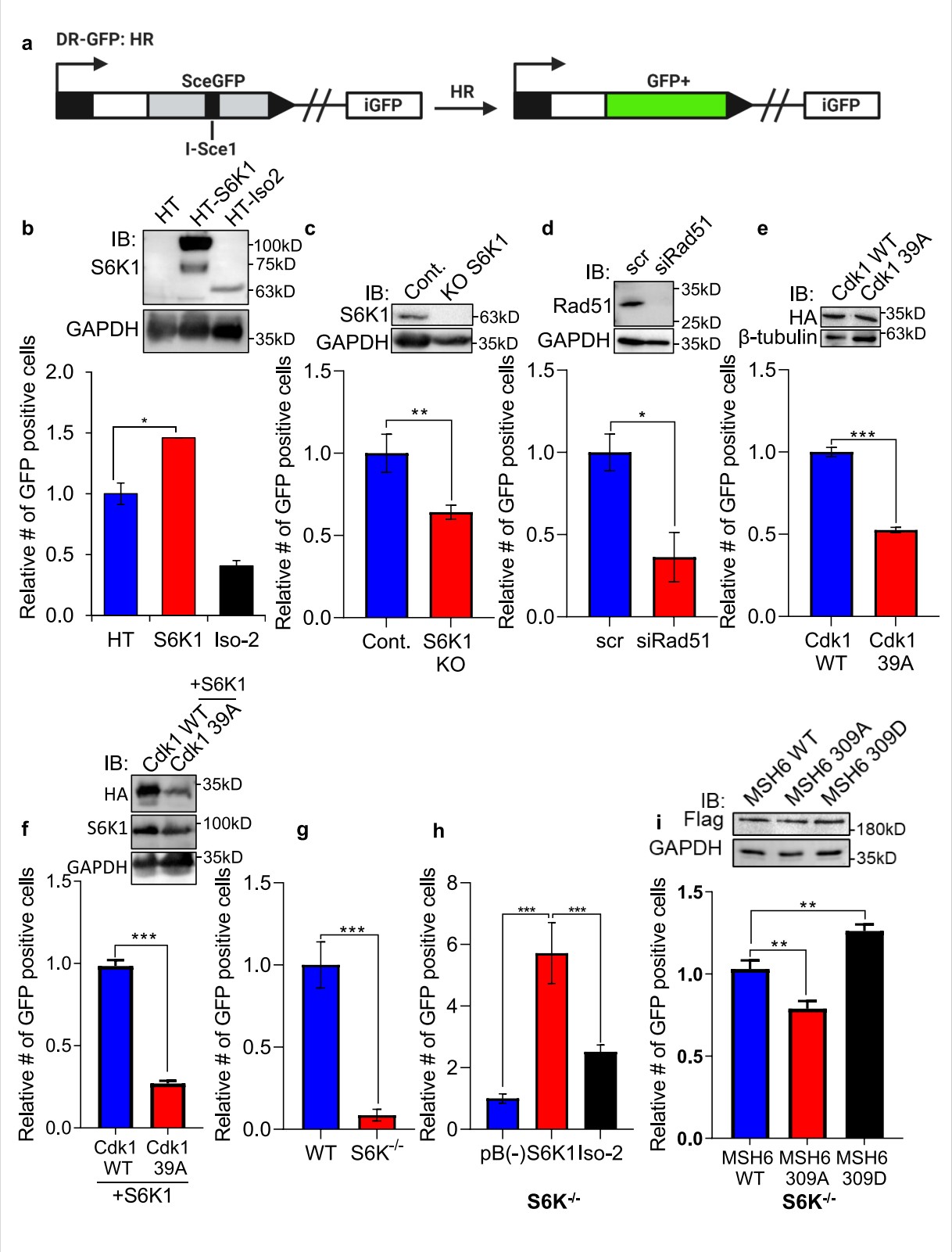

**Figure 6.** S6K1 overexpression enhances homologous recombination and mismatch repair. (**a**) Schematic diagram of DSB reporter plasmid. SceGFP is a GFP gene that contains an I-SceI endonuclease site within the coding region. Cleavage of the I-SceI site in vivo and repair by HDR (Homologous Directed Repair) coupled by the downstream iGFP repeat results in GFP positive cells. (**b**) U2OS cells expressing DSB reporter plasmid (U2OS-DR-GFP) were co-transfected with ISceI and either Halo-tag empty vector (HT), HT-S6K1 or HT-Iso2. Western blot showing S6K1 expression (top panel). Cells were

*Figure 6 continued on next page*

Figure 6 continued

analyzed by FACS for GFP expression (bottom panel). (**c**) U2OS-DR-GFP cells knocked-out for S6K1 using CRISPR were transfected with ISceI. Western blot showing S6K1 expression (top panel). Cells were analyzed by FACS for GFP expression (bottom panel). (**d**) U2OS-DR-GFP cells transfected with either control siRNA (scr) or siRNA specific for Rad51 (siRad51) were co-transfected with ISceI. Western blot showing Rad51 expression (top panel). Cells were analyzed by FACS for GFP expression (bottom panel). (**e**) U2OS-DR-GFP cells were co-transfected with ISceI and WT CDK1 or mutant CDK1 39 A. Western blot showing HA tag expression (top panel). Cells were analyzed by FACS for GFP expression (bottom panel). (**f**) U2OS-DR-GFP cells were co-transfected with ISceI and S6K1 with either WT CDK1 or mutant CDK1 39 A. Western blot showing HA tag expression (top panel). Cells were analyzed by FACS for GFP expression (bottom panel). (**g,h**) Mismatch repair assay using MEF WT (WT) and S6K-/- cells (**f**) or S6K-/- cells stably expressing either empty vector (pB(-)), S6K1 or Iso-2 (**g**) transfected with EGFP heteroduplex plasmid. Forty-eight hr after transfection, cells were analyzed by FACS for GFP expression. (**i**) S6K-/- cells were co-transfected with either Flag-wild type MSH6 (WT), Flag-MSH6 mutant 309 A (309 A) or Flag-MSH6 mutant 309D (309D) and EGFP heteroduplex plasmid. Western blot showing Flag-tagged expression (top panel). Forty-eight hr after transfection, cells were analyzed by FACS for GFP expression (bottom panel). Data represents means ± SD of biological triplicates. *p<0.05 values were calculated using Student's t-test (two-tailed).

The online version of this article includes the following source data and figure supplement(s) for figure 6:

**Source data 1.** Western blots of *Figure 6*.

**Figure supplement 1.** Cell cycle analysis of U2OS-DR-GFP cells.

1). One explanation for this low overlap is that we analyzed, in addition to active S6K1, the interaction with a kinase-dead short isoform Iso-2 (p31), which is a naturally occurring kinase-dead version of S6K1. We also identified interactions with S6K1 K123 >A, a kinase-dead version of S6K1, which is identical in size and sequence to S6K1 and differs only in a lysine to alanine (K123A) substitution (*Supplementary file 1*). However, these results were not included in our final analysis. Both of these kinase-dead versions bind many more proteins than S6K1, possibly due to repulsion of S6K1 interactors by the negative charge added following phosphorylation. PCNA, one of S6K1 interactors, is known for its role in orchestrating many processes such as replication, DNA repair, cell cycle control and apoptosis *Moldovan et al., 2007*; *Maga and Hubscher, 2003*. Within the MMR system, PCNA is known to interact with MSH2 and MSH6 (*Moldovan et al., 2007*; *De Biasio and Blanco, 2013*). We show that two of the S6K1 interacting proteins, Cdk1 (Cdc2) and MSH6, are directly phosphorylated by S6K1 (*Figure 3*). Here, we assign S6K1 an additional nuclear function, regulation of DNA repair.

In addition to the in vitro phosphorylation assay (*Figure 3a–b*), where S6K1 induced MSH6 phosphorylation, co-transfection of S6K1 with MSH6 enhanced its phosphorylation also in cells (*Figure 3c–d*). We also observed phosphorylation of Cdk1 upon co-transfection with S6K1 in cells, which was enhanced upon treatment of cells with UV radiation (*Figure 3e–f*).

## Cdk1 phosphorylation at serine 39 coordinates G2/M cell cycle arrest and HRR

We show that S6K1 promotes G2/M cell cycle arrest (*Figure 5*, *Figure 5—figure supplement 1a,b*) and increases HRR (*Figure 6*), the preferred error-free repair pathway in G2/M phase (*Ciccia and Elledge, 2010*; *Farmer et al., 2005*). The cell cycle is tightly regulated by Cdk-cyclin complexes. Specifically, the Cdk1-cyclinB complex controls the G2/M checkpoint (*Johnson and Jasin, 2000*). We further demonstrate that phosphorylation of Cdk1, by S6K1, correlates with G2/M cell cycle arrest following exposure IR (*Figure 5*). We provide the first evidence for Cdk1 phosphorylation on serine 39 by S6K1, and show that this phosphorylation enhances HRR, as its substitution to alanine inhibits HRR (*Figure 6*). Cdk1 is known to be phosphorylated on serine 39 by CK2 (*Yamane and Kinsella, 2005*), a vital player in DDR and cell survival (*Edelbrock et al., 2013*; *Christmann et al., 2002*; *St-Denis and Litchfield, 2009*), and this specific Cdk1 phosphorylation was also shown to be important for chemotherapy induced G2/M cell arrest (*Yamane and Kinsella, 2005*). According to our results, S6K1/S6K2 are not essential for the G2/M checkpoint (*Figure 5a–d*). However, S6K1 augments the G2/M checkpoint to enhance cell cycle arrest upon DNA damage (*Figure 5a–d*). This might be the mechanism by which S6K1 overexpression protects cells from DNA damage-induced cell death, while S6K1 knockout mice are viable with no obvious signs of DNA damage response problems. In addition, we do not exclude the fact that there are other kinases that phosphorylate Cdk1, such as CKII, and therefore it is reasonable that S6K-/- MEFs will still activate the G2/M checkpoint. The budding yeast Cdk1, Cdc28, has recently been shown to directly phosphorylate the essential homologous recombination repair proteins Rad51 and Rad52 (*Lim et al., 2020*). This connection, that probably occurs also in mammals,

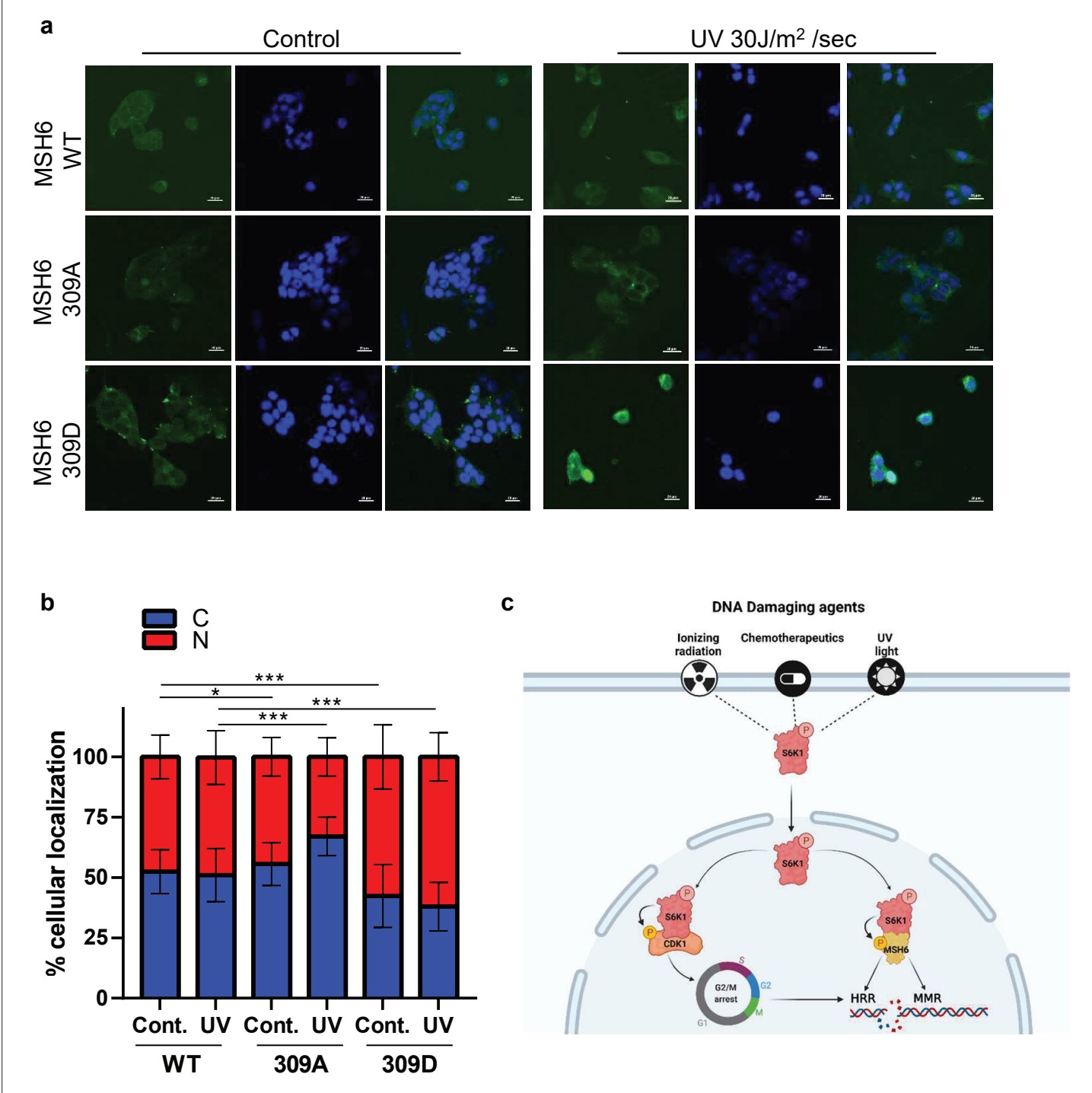

**Figure 7.** MSH6 309 phosphorylation is important for cellular localization. (**a,b**) MEF S6K -/- cells transfected with either Flag-MSH6 WT, MSH6 309 A or MSH6 309D were UV irradiated and fixed 2 hours later. Cells were stained using anti-Flag antibody and DAPI. Fluorescence was quantitated using NIS-element AR software. (**c**) Diagram depicting S6K1 entrance to the nucleus in response to DNA damaging agents. S6K1 phosphorylation of Cdk1 and MSH6 causes G2/M cell cycle arrest and affects DNA damage repair mechanisms.

strengthens the suggestion that modulation of Cdk1 activity regulates HR. It is important to mention that previous studies have shown that Cdk1 can phosphorylate S6K1, inhibiting its activity (*Kurabe et al., 2009*). This raises the possibility that there is a positive feedback loop between Cdk1 and S6K1.

## S6K1 enhances MMR by phosphorylation of MSH6 on serine 309

The MMR machinery signals directly to kinases such as ATR, CHK1 and CHK2 (*Wang and Qin, 2003*; *Adamson et al., 2005*; *Lord and Ashworth, 2012*), inducing several cascades in parallel, coordinating cell cycle checkpoints and DNA-repair activity (*Stojic et al., 2004*). MMR proteins not only signal to the repair mechanism but are also recruited by it; MSH2 and MSH6 have ATM and ATR phosphorylation motifs that undergo phosphorylation upon IR treatment (*Matsuoka et al., 2007*), and were shown to be recruited to sites of DNA damage upon UV irradiation (*Chou et al., 2010*). This evidence supports the existence of crosstalk between MMR proteins and DSB and SSB repair (*Ciccia and Elledge, 2010*) and illustrates MutSα involvement in UV induced DDR. Here, we show that S6K1 interacts with MSH6 and MSH2 (MutSα), upregulating their nuclear localization and elevating MMR activity (*Figures 2, 4 and 6* and *Figure 2—figure supplement 1*). The MSH6 NTR contains a PIP box, three identified Nuclear Localization Signals (NLS) (*Edelbrock et al., 2013*; *Gassman et al., 2011*) and a PWWP domain for binding to chromatin or chromatin associated proteins (such as histones). We show S6K1 phosphorylation of MSH6 in that region on multiple serines (*Figure 3* and *Supplementary file 4*). Some of these sites were reported to be mutated in familial colorectal cancer patients with germline MSH6 mutations and characterized by a low microsatellite instability phenotype. Moreover, point mutations in these residues were found to alter nuclear localization of MSH6 (*Gassman et al., 2011*; *Kolodner et al., 1999*). We show that S6K1 phosphorylates MSH6 on the serine cluster 252, 254, and 256, which is known to be phosphorylated also by CK2 (*Edelbrock et al., 2013*; *Gassman et al., 2011*; *Kaliyaperumal et al., 2011*) and serine 309 which is also phosphorylated by PKC (*Polo and Jackson, 2011*). To examine if serine 309 phosphorylation is directly involved in increased MMR activity observed in MEF S6K$^{-/-}$ cells overexpressing S6K1, we mutated this residue to either aspartate or alanine. Our results indicate that mutation of serine 309 to aspartate, which mimics the phosphorylated state, enhanced both MSH6 mediated repair activity (*Figure 6i*) and its nuclear localization (*Figure 7*). MutSα phosphorylation by CK2 and PKC, has already been shown to be involved in its nuclear translocation, corroborating our findings of increased nuclear transport of MutSα in S6K1 overexpressing MCF-10A cells and reduced nuclear localization following S6K1 knockdown (*Figure 4a–d*). Although we detected S6K1 translocation into the nucleus following DNA damage (*Figure 4e*), we cannot rule out that the phosphorylation of MSH6 and CDK1 by S6K1 occurs in the cytoplasm, as these proteins shuttle between the cytoplasm and the nucleus.

MMR-deficient cells are defective in cell cycle arrest in response to multiple types of DNA damaging agents *Li, 2008*. It is therefore reasonable to assume that S6K1 effects on cell cycle are mediated by both Cdk1 and MSH6 phosphorylation, two known proteins involved in MMR. S6K1 coordination of cell cycle arrest and DNA repair may also be attributed to phosphorylation of other proteins, in addition to MSH6 and Cdk1, which we identified as interacting with S6K1 but were not investigated further in this study. The fact that S6K1 protected from several types of DNA damaging agents suggests that, in addition to MMR and HHR, it might activate additional repair mechanisms. In conclusion, we show that S6K1 promotes both MMR and HRR activity (*Figures 6–7*) and both of these activities are dependent on S6K1 kinase activity (*Figure 6e and i*), revealing a novel role of S6K1 in DNA damage repair. These results suggest that activation of the mTOR pathway, following DNA damage, activates repair mechanisms through S6K1 kinase activity. Furthermore, it suggests that nutrient sensing is linked to the DNA repair machinery through the mTOR-S6K1 pathway (*Figure 7*).

## Clinical implications

It is becoming increasingly clear that cancer drug resistance is not only mediated by genes that drive or facilitate malignancy but also by genes that maintain essential functions required by both normal and malignant cells. DNA repair is one of the best examples of a process essential for both normal and cancer cells, and as such, it is an Achilles heel for cancer cells. For example, mutations in the tumor suppressors BRCA1 and BRCA2, on one hand, cause enhanced susceptibility to development of breast and ovarian cancers, but on the other hand, tumors in mutant BRCA1 and BRCA2 carriers

are extremely sensitive to DNA damaging drugs and to PARP inhibitors that eliminate their only functioning DNA repair arm (*Magesa et al., 2000*; *Yin and Shen, 2008*).

*RPS6KB1* gene amplification and overexpression is reported in about 8–10% of breast tumors and other cancers and correlates with poor prognosis (*Sinclair et al., 2003*; *Bostner et al., 2015*). Further clinical investigation is needed in order to establish if *RPS6KB1* amplification or activation is a predictive biomarker for intrinsic resistance to DNA damaging therapies such as chemotherapy and irradiation. Our results suggest that patients with *RPS6KB1* amplification will not benefit from chemotherapy or irradiation treatments. Alternatively, breast cancer patients with low *RPS6KB1* copy number might be hypersensitive to these treatments and will benefit the most. Moreover, our results predict that inhibition of S6K1 in patients with *RPS6KB1* amplification might enhance their response to chemotherapy. Several reports support our findings demonstrating that genotoxic stress regulates mTORC1, and that blockage of the Akt-mTORC1-S6K1 axis sensitizes cancer cells and tumors to radiation or chemotherapy (*Braunstein et al., 2009*; *Alexander et al., 2010*; *Albert et al., 2006*; *Shen et al., 2013*). We previously showed that several tumor types overexpress short inactive isoforms of S6K1 at the expense of the long isoform (*Ben-Hur et al., 2013*; *Karni et al., 2007*). Since the short isoforms are kinase-dead and do not enhance DNA repair or G2/M arrest following DNA damage (*Figures 5 and 6*), we hypothesize that tumors that show elevated levels of short S6K1 isoforms and low levels of long S6K1 isoforms will be more sensitive to DNA damaging agents, and possibly will show synthetic lethality using chemotherapy with PARP inhibitors.

## Materials and methods
### Cell lines
MCF-10A(human breast epithelial cell line, ATCC) cells were grown in DMEM/F12 medium supplemented with 5% (v/v) horse serum, 50 ng/ml epidermal growth factor (EGF), 10 µg/ml insulin, 0.5 µg/ml hydrocortisone, 100 ng/ml cholera toxin, penicillin and streptomycin[81]. MCF-7 (human breast epithelial cell line,ATCC), HEK293 (Human embryonal kidney cells, ATCC), HeLa(human cervix adenocarcinoma,ATCC), U2OS (human bone cell line, Prof. Michal Goldberg), MEF WT (mouse embryonic fibroblast, ATCC) and MEF S6K$^{-/-}$ cells were grown in DMEM with 10% (v/v) fetal calf serum (FCS), penicillin and streptomycin. MEF S6K$^{-/-}$ cells were described previously (*Pende et al., 2004*; *Pierce et al., 1999*). BTB474 (human breast Ductal Carcinoma cell line, ATCC), SKBR3 (human breast Adenocarcinoma cell line, ATCC) and HCC70 (human breast Ductal Carcinom cell line, ATCC) cells were grown in RPMI with 10% (v/v) fetal calf serum (FCS), penicillin and streptomycin. MEF S6K$^{-/-}$ and MCF-10A cells were transduced with pBABE-puro retroviral vectors encoding T7-tagged S6K1 isoforms as described (*Ben-Hur et al., 2013*). MLP-puro-shRNAs vectors, MCF-10A cell transductants were selected with puromycin (2 µg/ml) as described (*Ben-Hur et al., 2013*). The cell lines tested negative for mycoplasma contamination. Cells were authenticated by STR profiling.

### Halo tag cloning and pull down assay
Primers were designed using the Flexi Vector primer design tool http://www.promega.com/resources/tools/flexi-vector-primer-design-tool/ and cloning was performed using Flexi cloning protocol (Promega). Briefly, using directional cloning with SgfI at the N terminal and PmeI at the C terminal, the PCR products were subcloned into SgfI and PmeI digested Flexi acceptor vector (pFN21A). Halo Tag empty vector (G6591) was used as a control. Pull down was performed as described in the Halo tag technical manual TM342 (Promega). Briefly, HEK293 or HeLa cells were plated at a density of ~2.5–3 x 10$^6$ cells per 100 mm plate (70% confluency). The next day cells were transfected with 19–24 µg of Halo tag-fused S6K1 isoform plasmids using linear PEI. Forty-eight hr post transfection cells were washed with 10 ml cold PBS, manually scraped with 10 ml cold PBS, centrifuged 14,000 g at 4 C° for 10 minutes and the cell pellet was frozen overnight at –80 C°. Pellets were thawed with lysis buffer (50 mM Tris pH 7.5, 150 mM NaCl, 1% Triton X100, 1 mg/ml Na-deoxycholate) supplemented with protease inhibitor (Promega cat #G6521). The bait is covalently bound through HaloTag to the beads and therefore cannot be detected by blotting.

## Mass spectrometry

Cell pellets were thawed in Lysis buffer (50 mM Tris pH 7.5, 150 mM NaCl, 1% NP-40 supplemented with protease inhibitor cocktail (Promega cat #G6521)) and incubated on ice for 15 min. Lysates were passed through a 25 gauge needle and centrifuged at 14,000 g for 5 min at 4 °C. Clear lysates were transferred to a fresh tube containing 200 µl of HaloLink resin and binding was performed for 1 hr at room temperature. The beads were centrifuged at 800 g for 3 minutes and washed three times with Wash buffer 1 (0.1M Tris-HCl, 150 mM NaCl and 0.05% NP-40) and then three times with Wash buffer 2 (0.1M Tris-HCl, 150 mM NaCl). Laemmli buffer was added and samples were separated using SDS-PAGE. After staining of the acrylamide gel with GelCode blue (Thermo Fisher) each lane was cut into 3 pieces according to protein size and analyzed separately by LC-MS/MS. Alternatively, on bead digestion was performed by incubating beads (HaloLink resin) with 0.4 µg sequencing grade modified trypsin in 2 M urea and 1 mM DTT for 2 hr at room temperature. Following eluate collection, beads were washed again with 5 mM iodoacetamide in 2 M urea and collected to the same tubes. Eluates were incubated overnight at room temperature to complete digestion. Peptides were then purified on C18 StageTips. LC-MS/MS analysis was performed on the EASY-nLC1000 UHPLC coupled to the Q-Exactive mass spectrometer through an EASY-Spray ionization source (Thermo Scientific). Each sample was analyzed with a 4-hr gradient using a 50 cm EASY-Spray column (Thermo Scientific; Dionex). Raw MS/MS files were analyzed with the MaxQuant software, the Andromeda search engine and the label-free quantification (LFQ) algorithm. To extract proteins with significantly differential binding, we compared the LFQ-intensities of the various samples using t-tests between each sample relative to its control (Halo Tag empty) (permutation-based FDR <0.05).

## Treatment with chemotherapeutic agents, UV irradiation, and γ irradiation

One day after cells were seeded the plates were washed once with PBS, the PBS was removed and cells were irradiated as described in the figure legends (6.5–30 $J/m^2/sec$ UVC or 1–5 Gy γ irradiation) and harvested as indicated. For treatment with chemotherapeutic agents cells were treated with either Doxorubicin 1–6 mg/ml (Sigma, cat #D1515), NCS 1–150 ng/ml (Sigma, cat #N9162,) or CDDP 10 µM (Sigma, cat #C2210000) as described in the figure legends.

## Immunoblotting

Cells were lysed in Laemmli buffer (10% glycerol, 0.05 M Tris pH 6.8, 5% β-mercaptoethanol, 3% SDS) at 90 C°. Primary antibodies used - **Santa Cruz:** β-actin (dilution 1:200, cat #1616), GAPDH (dilution 1:1000, cat #25778), MSH2 (dilution 1:200, cat #365052), HA (dilution 1:1000, cat #7392). **Novagen:** T7 tag (dilution 1:5000, cat #69522–4). BD Transduction Laboratories: Total S6K1-anti-p70 (dilution 1:250, cat #611261), MSH6 (dilution 1:250, cat #610919). **Cell Signaling Technologies:** phospho-S6K1 Thr389 (dilution 1:1000, cat #9205), p-S6 ser240/244 (dilution 1:1000, cat #2215), p-S6 ser235/236 (dilution 1:1000, cat #2211), total S6 (dilution 1:1000, cat #2217), PCNA (dilution 1:500, cat #2586), phospho-Akt substrate(RXRXXS/T) (dilution 1:500, cat #9614), cleaved caspase 3 (dilution 1:500, cat #9661), mTOR (dilution 1:1000, cat #2972), GFP (dilution 1:500, cat #2956), Cyclin A2 (dilution 1:1000, cat #4656), PARP1 (dilution 1:1000, cat #9542), phospho-histone H2A.X (Ser 139) (dilution 1:1000, cat # 9718), p-ATM(ser1981)(dilution 1:1000, cat #5883), ATM (dilution 1:1000, cat #2873). **Promega:** Halo Tag pAb (dilution 1:2000, cat #G9281). Sigma: β tubulin (dilution 1:1000, cat #T8535), β-catenin (dilution 1:2000, cat #C7207). **Millipore:** MCM7 (dilution 1:1000, cat #MABE 188), p-CDK1 (dilution 1:900, cat #MABE 229). **Abcam:** CDK1(dilution 1:1000, cat #ab18), Caspase 2 (dilution 1:1000, cat #ab179520). **Affinity biosciences:** p-ATM(ser1987) (dilution 1:1000, cat #AF8410). SRSF1 (dilution 1:100, mAb AK96 culture supernatant). Secondary antibodies used were from Jackson ImmunoResearch (dilution 1:1000) Peroxidase AffiniPure Goat Anti-Mouse IgG (H+L) (cat #115-035-003), Peroxidase-AffiniPure Goat Anti-Rabbit IgG (cat #111-035-003) or Peroxidase-AffiniPure Donkey Anti-Goat IgG (H+L) (cat #705-035-003).

## Immunoprecipitation (IP)

HEK293 or HeLa cells were transfected with 24 µg plasmid DNA/100 mm plate of either Halo tag-S6K1 or Iso-2 plasmids and lysed 48 hours later in CHAPS buffer [84]. A total of 800–1500 µg of total protein lysate was incubated overnight with 1 µg of either anti-PCNA, CDK1 or MSH6 antibody bound to 40 µl

of 50% protein A+G sepharose beads or G-sepharose beads. After washing four times with CHAPS buffer, beads were incubated with 50 µl of 2 X Laemmli buffer, boiled and separated by SDS-PAGE. Secondary antibodies for immunoblotting after IP were: Rabbit true-blot (Rockland Laboratories, cat #18-88166-31) or mouse true-blot (Rockland Laboratories, cat #18-8817-31). These antibodies were used to avoid nonspecific bands caused by the heavy and light chain of the primary antibody. For IP of FLAG-MSH6, anti-FLAG beads were used (Sigma, cat #A2220).

## Fluorescence microscopy

Cells were plated onto 12 well plate on coverslips. Halo tag TMR Direct Ligand (Promega, cat # G2991) was used for staining of Halo tag transfected fixed cells, according to manufacturer's protocol. Cells were co-stained using anti-phospho-histone H2A.X (Ser 139) (Cell Signaling, cat # 9718) and secondary antibody Goat anti-Rabbit IgG (H+L) Highly Cross-Adsorbed Secondary Antibody, Alexa Fluor 488 (ThermoFisher #A11034). Cells were scored for 'nuclear staining' manually. Quantification was performed using NIS-Elements AR software.

## Clonogenic assay

MEF WT or MEF S6K$^{-/-}$ cells transduced with retroviruses encoding for S6K1 isoforms were seeded 800 cells/well in a six-well plate in triplicate. The next day, the cells were irradiated with either UV (6.5 J/m$^2$/sec) or γ irradiation (1–5 Gy). After 14 days, colonies were fixed and stained with methylene blue as described previously (*Karni et al., 2007*) and the number of colonies in each well was calculated.

## Trypan-blue exclusion assay

Cells were seeded 50,000 cells per well in a 12-well plate in triplicate. The following day media was replaced with media containing either 2–8 µg/ml of doxorubicin, 5 mM of CDDP or DMSO control or irradiated with either UV (8 J/m$^2$/sec) or γ irradiation (5 Gy). After 48 hr, cells were trypsinized and spun down. Media and PBS wash were also collected. Cells were resuspended in HBSS and the percentage of dead cells was determined by 0.4% trypan blue staining and BioRad cell counter.

## Subcellular fractionation assay

MCF-10A cells transduced with viruses expressing either empty vector (EV), S6K1, Iso-2 or S6K1 specific shRNA were seeded (5x10$^5$ cells per 60 cm plate). Twenty-four hs later, the cells were subjected to UV (30 J/m2/s) and after 2 hr trypsinized, washed with cold PBS and centrifuged (1500 rpm for 5 min). The cells were resuspended in CE buffer (Hepes 10 mM, MgCl$_2$ 1.5 mM, KCl 10 mM) and incubated for 5 min on ice. Subsequently, cells were resuspended in CE buffer containing 0.2% NP-40, incubated for 5 min on ice and centrifuged at 4 °C (6500 rpm for 3 min). The supernatant was collected (CE) and the pellets were resuspended in NE buffer (Hepes 20 mM, MgCl$_2$ 1.5 mM, NaCl 420 mM, EDTA 0.2 mM and 25% glycerol). Next the tubes containing NE buffer were vortexed at maximal speed for 1 min and subjected to three freeze/thaw cycles. The supernatant was collected (NE). Both CE and NE lysates were analyzed by immunoblotting using the designated antibodies. SRSF1 and tubulin served as nuclear and cytoplasmic markers, respectively.

## In vitro kinase assay

In vitro kinase assay was performed using recombinant proteins; S6K1 (University of Dundee, cat # DU784), S6 (Abnova, cat #H00006194-P01), CDK1 (Prospec, cat# pka-076) and MSH6 (aa1-718, as described in protein purification section). In brief, 300 ng of recombinant S6K1 was incubated alone or with 5 µg of either recombinant S6, CDK1 or MSH6 in 20 µl reaction buffer containing 20 µM cold ATP, 0.5 µCi γ-$^{32}$P-ATP, 30 mM MgCl2, 10 mM HEPES (pH 7.5), 50 mM EGTA, 10 mM β-glycerophosphate, 5 mM NaVO4, 50 mM β-mercaptoethanol, and 0.5 mM dithiothreitol (DTT). Reactions were shaken (1200 rpm) for 1 hr at 30 °C and neutralized by the addition of 18 µl 4×Laemmli buffer and heated for 5 min at 95 °C. Thirty µl of the final volume was separated by SDS-PAGE, transferred to nitrocellulose by western blotting and exposed to Typhoon FLA 7000 for detection of radioactive bands. After exposure, the membrane was probed with the indicated antibodies to visualize the recombinant proteins.

## Protein purification

Human *MSH6* (aa 1–718) was cloned into pET-32a vector using primers: Forward 5'-GGCGCCAT GGGATCCTCGCGACAGAGCACCCTGTA-3' and Reverse 5'-GATTCAAA GCGGCCGCCTAGCTGA

CTGTGTCAGAATCCA-3'. The proteins were expressed in *Escherichia coli* T7 Express (New England Biolabs) and grown in Luria-Bertani medium. Cultures were inoculated using 1% (v/v) of a saturated overnight culture and were grown at 37 °C to $OD_{600}$ 0.4–0.6. Proteins were induced at 16 °C overnight by addition of 150 µM isopropyl-β-D-thiogalactoside. Cells were harvested by centrifugation (6200×g, 15 min) and stored at −80 °C for later use. Proteins were purified by resuspending cell pellets in lysis buffer (50 mM sodium phosphate pH 8.0, 300 mM NaCl, 15 mM imidazole, and 5% glycerol) after adding 1 mM phenyl-methyl sulphonyl fluoride (PMSF). Cells were disrupted using a microfluidizer (Microfluidics), and the lysate was centrifuged at 68,900 g for 1 hr to remove cell debris. The lysates was subjected to Nickel gravity column using 5 ml His-Trap columns (GE Healthcare), and protein was eluted with 300 mM imidazole in column volumes. Fractions containing purified protein were pooled and dialyzed (20 mM Tris pH 7.8, 100 mM NaCl, and 5 mM β-mercaptoethanol) overnight at 4 °C. Fractions containing purified protein were concentrated and flash-frozen in liquid $N_2$ before being stored at −80 °C.

## Flow cytometry

For analysis of GFP and mCherry expression cells were harvested and centrifuged with PBS for 5 min at 1500×g. The pellets were resuspended in 1% BSA in PBS and subjected to flow cytometry (LSR-II Analyzer). Typically, 50,000 cells were counted.

For cell cycle analysis cells were harvested and centrifuged with cold PBS for 5 min at 1500×g. The cells were fixed with cold 100% ethanol and stored at −20 °C for several days. The cell pellets were incubated with RNase A 10 µg/ml (Sigma, cat #R5125) and stained with propidium iodide(PI) 5 µg/µl (Sigma, cat #P4170) and subjected to flow cytometry (LSR-II Analyzer). The analysis was performed using the FCS express 4 Flow cytometry software. For cyclin analysis cells were fixated using 1% PFA for 10 min in room temperature, then were permobilized using cold 100% methanol and stored in −20 °C for several days. The cell pellets were incubated for 30 min with PE anti-Cyclin A (1:500, BioLegend Cat: 644044), Alexa 488 anti pH3 (1:500, BioLegend Cat:650803) or Alexa 647 anti-Cyclin B1 (1:200, BioLegend Cat: 647906). DNA content were stained using DAPI.

## Site-specific mutagenesis

The human *CDK1* S39A mutation was introduced using Quickchange (Agilent, cat #600670) using specific primers: Forward 5'-AAAATCAGACTAGAAGCTGAAGAGGAAGGGGTT-3' and reverse 5'-AACCCCTTCCTCTTCAGCTTCTAGTCTGATTTT-3'. The mutated plasmids were confirmed by DNA sequencing of coding regions.

## Mismatch repair assay

The mismatch repair assay was performed as described previously (*Zhou et al., 2009*) using plasmids pGEM5Z (+)-EGFP (Addgene, cat #65206) and p189 (Addgene, cat #65207). Cells were transfected using TransIT-X2 transfection reagent (Mirus, cat# MIR6000). In order to control for transfection efficiency cells were co-transfected with a plasmid expressing mCherry (pmCherry-C1). The number of GFP positive cells in each sample was divided by the number of mCherry positive cells. GFP and mCherry expression were measured using flow cytometry (LSR-II Analyzer).

## Homologous recombination assay

The assay was performed as described previously using U2OS-DR-GFP cells and ISceI plasmid (*Shahar et al., 2012*). The ISceI plasmid, S6K1, Iso-2 and Cdk1 and its mutant plasmids were transfected into the cells using TransIT-X2 transfection reagent (Mirus, cat# MIR6000). siRNA oligonucleotides specific for Rad 51 were purchased from Sigma (ESIRNA HUMAN RAD51 EHU045521). The siRNA was transfected into cells using Lipofectamine 2000 (Invitrogen, cat #11668019). CRISPR/Cas9 system was used for knockout of S6K1 in U2OS-DR-GFP cells. The target gRNA oligonucleotide was subcloned into the pLenti CRISPR V2 plasmid and cells were transduced with lentivirus. The oligonucleotide sequence for the S6K1 guide (sgRPS6KB1) is 5'-CCATGAGGCGACGAAGGAGG-3'. In order to control for transfection efficiency we co-transfected with a plasmid expressing mCherry (pmCherry-C1). The number of GFP positive cells in each sample was divided by the number of mCherry positive cells. GFP and mCherry expression were measured using flow cytometry (LSR-II Analyzer).

## Kaplan-Meier plot

The KM plots for breast cancer were created using Kaplan-Meier Plotter (kmplot.com) for the RPS6KB1 gene, using three Affymetrix probe sets (ID: 211578, 226660, 204171). The 'no chemotherapy' graph was created using 'systemically untreated patients' cohort category and the 'chemotherapy treated' graph was created using 'patients with the following systemic treatment' cohort category.

## Statistical analysis

All data presented as histograms refer to a mean value ± SD of the total number (n=3–6) of independent biological experiments. An unpaired, two-tailed t test was used to determine p values.

## Acknowledgements

The authors wish to thank Prof. Michal Goldberg (HUJI) for the U2OS-DR-GFP cells and ISceI plasmid; Prof. Xiaohong (Mary) Zhang (Karmanos Cancer Institute, Detroit) for MSH2 and MSH6 plasmids; Prof. Nabieh Ayoub (Technion, Israel Institute of Technology) for MSH6 plasmids and Prof. Michael Brandeis (HUJI) for CDK1 plasmids and CDK1 antibody. Funding information This study was supported by an ISF Grant 1290/12 and ISF Grant 1510/17 (to RK), Israel Cancer Association grant (to RK) and ERC grant (to RIA). Figures were created with https://biorender.com/.

## Additional information

### Funding

| Funder | Grant reference number | Author |
|---|---|---|
| Israel Science Foundation | 1290/12 | Rotem Karni |
| Israel Science Foundation | 1501/17 | Rotem Karni |
| Israel Cancer Association | 20181132 | Rotem Karni |
| H2020 European Research Council | 682118 | Rotem Karni |

The funders had no role in study design, data collection and interpretation, or the decision to submit the work for publication.

### Author contributions

Adi Amar-Schwartz, Conceptualization, Software, Formal analysis, Supervision, Validation, Investigation, Visualization, Methodology, Writing – original draft, Project administration, Writing – review and editing; Vered Ben Hur, Conceptualization, Data curation, Formal analysis, Validation, Investigation, Visualization, Methodology, Writing – original draft, Project administration, Writing – review and editing; Amina Jbara, Conceptualization, Data curation, Formal analysis, Validation, Investigation, Visualization, Methodology; Yuval Cohen, Validation, Investigation, Visualization, Methodology; Georgina D Barnabas, Bayan Mashahreh, Fouad Hassouna, Mohammad Abu-Odeh, Rami Aqeilan, Resources, Methodology; Eliran Arbib, Methodology; Zahava Siegfried, Conceptualization, Writing – original draft, Writing – review and editing; Asaf Shilo, Validation, Investigation; Michael Berger, Resources, Software, Investigation, Methodology; Reuven Wiener, Resources, Investigation, Methodology; Tamar Geiger, Resources, Software, Methodology; Rotem Karni, Conceptualization, Resources, Formal analysis, Supervision, Funding acquisition, Methodology, Writing – original draft, Project administration, Writing – review and editing

### Author ORCIDs

Zahava Siegfried http://orcid.org/0000-0001-9649-7434
Bayan Mashahreh http://orcid.org/0000-0002-1667-1322
Michael Berger http://orcid.org/0000-0002-3469-0076
Rami Aqeilan http://orcid.org/0000-0002-6034-023X
Tamar Geiger http://orcid.org/0000-0002-9526-197X
Rotem Karni http://orcid.org/0000-0002-7552-9617

Decision letter and Author response

Decision letter https://doi.org/10.7554/eLife.79128.sa1

Author response https://doi.org/10.7554/eLife.79128.sa2

## Additional files

### Supplementary files

• Supplementary file 1. Mass Spectrometry. MS results of pull-downs performed on HEK293 cells transfected with Halo-Tag (HT)-S6K1 isoforms as described in section meterials and Methods. Proteins that were found bound to either of the (HT)-S6K1 isoforms by both methods are highlighted in yellow (total of 164 matches).

• Supplementary file 2. Shared proteins from Venn diagrams.

• Supplementary file 3. Predicted S6K1 phosphorylation sites. Possible RX(RXXS) S6K1 phosphorylation motifs on MSH2, MSH6 and CDK1 identified by phosphonet; http://www.phosphonet.ca/ (RXXS), marked in gray, represents the core motif shared by all S6K1 substrates. X, any amino acid, R, Argenine and S, Serine. The putative phosphorylated serine is highlighted in red.

• Supplementary file 4. Phosphorylation sites in vitro kinase assay.

• MDAR checklist

### Data availability

All data generated or analysed during this study are included in the manuscript and supporting file. Source data files have been provided for Figures 1–4, 6, and Figure 1—figure supplements 1–3, Figure 2—figure supplement 1 and Figure 4—figure supplement 1.

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
