## [Editor Report]

This study provides evidence for a novel function of S6K1 in DNA repair. The findings explain how amplification of RPS6KB1 gene, which encodes the S6K1 protein, in breast cancers is associated with a poor outcome after chemotherapy. The conclusions of the paper are well-supported by the experimental results. The proteomics data leading to the discovery of Cdk1 and Msh6 as targets of S6K1 are novel and important for the cancer research field.

---

## [Decision Letter]

[Editors' note: this paper was reviewed by Review Commons.]

---

## [Author Response]

We are happy to submit our revised manuscript “S6K1 phosphorylates Cdk1 and MSH6 to regulate DNA repair” to *eLife*. We thank the Reviewers for their helpful comments, which we have considered carefully. We note that the reviewers appreciated the importance of this study as seen by their comments "This study provides evidence for a novel function of S6K1 in DNA repair. These observations may help understand how amplification of RPS6KB1 gene that encodes S6K1 in breast cancers is associated with poor chemotherapy outcome" and "This study adds value to our understanding of DNA damage responses in cancer, particularly with regards to a potential role for the mTOR signaling pathway in regulating this process". We have now clarified the points that were not clear to the reviewers in the original manuscript. In addition, we have included many additional experiments and controls that we believe strengthen our conclusions.

Our point-by-point response is included below.

Reviewer #1.Specific comments:1. The way the IP data was presented is strangely inconsistent and confusing. For instance, In Figure 2C and 2f, blotting of the bait was not shown.

The HT pull-down is based on covalent binding of the Halotag to the beads and thus, after binding of the Halotag-conjugated protein, it cannot be released from the beads and only the proteins which were copulled down with it can be separated on the SDS-PAGE. This is the reason for showing flow-through (unbound) as a control since the levels of the pulled down Halotag-conjugated protein cannot be detected in the Western blot. We have now clarified the design of the HaloTag experiments explaining why blotting of HT is not shown in Figure 2c and 2f (Material and methods section).

They chose to show flow through for figure 2d and 2e but not for other panels.

We apologize for this and now include flow through for all panels.

The lysate and FT was shown in separate panels (2d and 2e) for HeLa and HEK293 cells but pulldown for the two cells were grouped into one panel (2f). For IP experiment in Figure 2g and 2h, HT antibody was used for lysate, but S6K1 and phospho-S6K1 antibodies were used for pulldown products.

Please see explanation to specific comment 1. In these experiments the bait is covalently bound through HaloTag to the beads and therefore cannot be detected by blotting.

There is no input blot for figure 2i.

We apologize for this and have now included input blot for Figure 2i.

No reasonable explanations have been provided for these inconsistencies or choices. I also strongly suggest that the authors group blots for each IP experiment into one figure panel that includes both input and IP samples.

We have now included blots that the reviewer requested and rearranged the panels to include both input and IP samples for each experiment in the same panel (see new Figure 2).

2. "Immunoblot of PCNA immunoprecipitated lysates showed interaction of PCNA with both S6K1 and Iso2 in HEK293 cells (Figure 2g,h) and in HeLa cells (Supplementary Figure 4f,g)." I do not see any interaction between S6K1 and PCNA in Supplementary Figure 4g.

The interaction of S6K1 with PCNA (and other substrates) may be difficult to capture due to possible repulsion of S6K1 interactors by the negative charge added following phosphorylation. This common phenomenon in kinase-substrate interactions is explained on page 6 of manuscript "As phosphorylation usually causes conformational changes of the substrate due to steric repulsion of the negatively charged phosphate groups, we hypothesized that the S6K1 kinase inactive variant^7^, short isoform Iso-2, may have a stronger interaction with all putative substrates". This is now clarified again in the revised manuscript on page 7 "It should be noted that the interaction with Iso-2 is stronger due to lack of steric repulsion caused by S6K1 phosphorylation".

3. Labeling in Figure 3a is confusing. Also, CDK1 was blotted for but MSH6 was not. Why was S6K1 seen in CDK1 WB?

We agree that Figure 3a is complicated and a bit confusing. S6K1 auto-phosphorylates in the presence of ATP, so in every P^32^ exposure it is detected. In the WB of CDK1+S6K1 (right side of panel 3a) S6K1 is the recombinant S6K1 that was added to the in-vitro kinase assay. MSH6 protein levels were hard to detect in Figure 3a. However, in the similar “cold” in vitro kinase assay experiment in Figure 3b (bottom right), it is clearly detected and the assay is similar, apart from the detection method of the phosphorylation.

4. "Cdk1 and MSH6, but not PCNA, have multiple putative S6K1 phosphorylation sites (Supplementary Table 3). " The authors have not reasoned why they only chose to examine these three proteins.

The reviewer is correct, Cdk1 has two putative S6K1 phosphorylation sites and MSH6 has 12 putative sites. We chose to focus on these three proteins due to their roles in cell cycle regulation (Cdk1), DNA damage repair (MSH6) and DNA synthesis and repair (PCNA). These are well studied proteins with reagents available for performing the experiments (antibodies). Clearly there are other proteins that were identified that interact with S6K1 and these proteins can be the focus of future studies.

5. "To eliminate the possibility of the presence of a contaminating protein, which may be acting as a kinase or substrate in our assay, we performed the in vitro kinase assay using recombinant Uba5 as a negative control. Uba5 does not contain any RXXS phosphorylation motifs but does have 26 serine and 19 threonine residues. In our assay S6K1 did not phosphorylate Uba5 (Supplementary Figure 5a,b)." This is a worthwhile control experiment to have. But it is not to address the possibility that the author raised about a contaminating kinase. One way to address this question would be to use a kinase-dead S6K1.

We agree with the reviewer that the use of Uba5 as a negative control is worthwhile. The Uba5 control shows that there is no contaminating protein that is acting as a kinase per se (able to phosphorylate Uba5 multiple serines and threonines) in our in vitro kinase assay. Unfortunately, there is no commercial kinase dead S6K1 available and we were not able to purify recombinant kinase-dead S6K1 protein (due to aggregation and inclusion body issues), therefore we were not able to perform in vitro kinase assay with kinase-dead S6K1. However, in the experiments in cell culture, the kinase-dead version – Iso-2 – was unable to enhance MSH6 phosphorylation (Figure 3c-d).

6. The authors made an interesting finding that in cells over-expressing S6K1, γ-H2AX was largely suppressed. They attributed this to enhanced repair in S6K1 expressing cells. In this experiment, the cells were fixed and stained just 2 hours after UV irradiation. UV irradiation mostly causes pyrimidine dimers and photo adducts, which need to be converted to DNA breaks during DNA replication or other DNA transactions to induce γ-H2AX. The time window is simply too short to allow all these repair processes to complete. In contrast, the cell cycle effect was observed 24 hours after UV or IR treatment in Figure 5.Therefore, it is more likely DNA damage signaling is suppressed in these cells that lead to lower gammaH2AX level. If the authors wants to support their conclusion, they need to at least conduct a time course experiment to show that the initial γ-H2AX signaling was not affected in these cells.

We agree with the reviewer that a time course experiment would clarify whether γ-H2AX signaling is affected in S6K1 over-expressing cells. We thank the reviewer for suggesting this experiment. We have now performed a time course experiment on WT MEF cells and S6K-/- MEF cells from 5 min to 8 hours after UV irradiation. We found that γ-H2AX phosphorylation was detected 1 hour after UV irradiation. However, we did not observe any significant difference in γ-H2AX signaling between WT and S6K-/- MEF cells (Author response image 1, panel a). We also performed a time course experiment on U20S cells transfected with either S6K1 or Iso-2 after UV irradiation. Similarly, γ-H2AX phosphorylation was detected 1 hour after UV irradiation but we did not observe any significant difference in γ-H2AX levels between endogenous S6K1 (HT), WT HT-S6K or HT-Iso-2 (Author response image 1, panel b). These results confirm that the initial γ-H2AX signaling was not affected in these cells.

**Author response image 1. sa2fig1:** Time Course of pH2AX levels. (a) MEF WT and S6K-/- cells were irradiated with UV30J/m2/sec. Cells were harvested at different time points after irradiation. Western blot analysis shows pH2AX levels. (b) U2OS cells overexpressing either HT, HT-S6K1 or HT-iso-2 were irradiated withUV 30J/m2/sec. Cells were harvested at different time points after irradiation. Western blot analysis shows pH2AX levels.

7. The increased HR efficiency in S6K1-expressing cells is expected, given that homologous recombination mainly occurs in G2/M phase. What is interesting is the decreased HR efficiency in S6K1 knockout cells and in Cdk1-39A cells. But since the CDK1 experiment was done by over-expressing WT or 39A mutant Cdk1, it is difficult to conclude what is the effect of 39A in normal cells. It is possible that Cdk1 over-expression changes cell cycle profile, which further complicate the interpretation. There is also not enough evidence that S6K1 promotes G2M arrest and HR vis CDK1. To prove this, the authors need to introduce 39A mutation to native Cdk1 and test if this mutant abolishes the effect of S6K1 expression.

We agree with the reviewer that it is possible that Cdk1 overexpression could possibly change the cell cycle profile, further complicating interpretation. The reviewer suggested that we introduce 39A mutation to native Cdk1 and test if this mutant abolishes the effect of S6K1. However, we chose to perform cell cycle experiments in U20S-DR-GFP cells with and without overexpression of S6K1 co-transfected with either Cdk1 WT or Cdk1 39A mutations. These experiments were more feasible for us to complete rather than introducing 39A mutation to native Cdk1. We have now included cell cycle analysis using U2OS-DR-GFP cells either overexpressing S6K1 or Iso-2; or overexpressing Cdk1 WT or Cdk1 39A; or overexpressing S6K1 with either Cdk1 WT or Cdk1 39A. Note that our cell cycle analysis is much more inclusive now as we separated the cell cycle by adding markers for -late ‘S’ phase – cyclin A, ‘G2’-high cyclin A+B and ‘M’ phase- p-H3. We did not observe any difference in cell cycle in the cells with either Cdk1 WT or Cdk1 39A (new Figure 6—figure supplement 1). We therefore can conclude that Cdk1 overexpression does not change the cell cycle profile in these cells.

We also tested whether introduction of Cdk1 39A abolishes the effect of S6K1 on homologous recombination. U2OS-DR-GFP cells transfected with S6K1 and Cdk1 39A showed less homologous recombination than U2OS-DR-GFP cells transfected with S6K1 and Cdk1 WT, supporting the conclusion that S6K1 promotes HR via Cdk1 (new Figure 6f).

Reviewer #2.Major critiques:1) Figure 4 studies show an increase in nuclear Msh6 upon overexpression of S6K1, but not its kinase-dead isoform Iso-2. It is unclear whether this is due to phosphorylation of Msh6 by S6K1 that regulates nuclear/cytoplasmic shuttling, or through a separate mechanism of action. Are these differences in cytoplasmic/nuclear ratio also observed with an Msh6 expression construct with putative S6K1 phosphorylation sites mutated?

We thank the reviewer for his/her suggestion.

We performed IF on MEF cells expressing either WT-MSH6, 309A or 309D. After UV damage we observed shuttling of MSH6 309A to the cytoplasm, in contrast to MSH6 309D which is more nuclear also compared to WT-MSH6. Importantly, in these cells and using the same constructs we observed differential NER activity (Figure 6i), connecting the differential localization to the function (Figure 7a,b).

Furthermore, is there a difference in Msh6 mRNA levels?

We did not detect any difference in mRNA levels of Msh6 in MCF10A cells overexpressing either S6K1 or Iso-2 after UV irradiation (Author response image 2, panel a) or in MCF10A cells expressing S6K1 specific sh1 or sh2 after UV irradiation (Author response image 2, panel b).

**Author response image 2. sa2fig2:** MSH6 mRNA levels in cells after UV irradiation. a. MCF10A cells overexpressing either S6K1 or Iso-2 were either untreated or treated with UV 30J/m2/sec. Q-RT-PCR was used to quantitate MSH6 mRNA levels. b. MCF10A cells overexpressing S6K1 specific sh1 or sh2 were either untreated or treated with UV 30J/m2/sec. Q-RT-PCR was used to quantitate MSH6 mRNA levels.

Finally, the altered expression/stabilization of S6K1 after UV treatment in Figure 4f seems intriguing – are the cells with overexpressed S6K1 arrested in a particular cell cycle phase? The reduction in gH2ax could reflect enrichment in non-S phase cells upon S6K1 overexpression.

We presented experiments to determine if reduction in γ-H2AX could be due to changes in cell cycle upon S6K1 overexpression using U20S-DR-GFP cells which are U2OS cells containing a DR-GFP cassette. The results were shown in the original manuscript (supplementary Figure 7a-b). We have now repeated this and included the results in new Figure 6—figure supplement 1a-d. No change in cell cycle distribution was detected in U20S-DR-GFP cells with S6K1 overexpression compared to control cells (see also reply to Reviewer #1, comment 7).

2) The cell cycle phenotype of S6K1 overexpressing cells after IR in Figure 5c-d is striking. The authors should use a mitosis-specific marker such as phospho-HistoneH3-S10 in their flow cytometry analyses, in addition to DNA content dyes. The phenotype appears to be more of a reduction in S phase fraction, which should be more directly quantified by EdU/BrdU incorporation assays.

We thank the reviewer for this suggestion. We have now repeated the cell cycle experiments to include mitosis-specific marker (p-H3), late S phase marker (cyclin A) and G2 phase marker (cyclin B).

The use of the cyclin markers allow us to distinguish between G2 and M fractions based on cyclin levels as detected by FACS in MEF -/- cells stably expressing S6K1 or Iso-2: High Cyclin A, low cyclin B = S-phase; High Cyclin A+B = G2 phase; High p-H3 = M phase. After IR damage we observed an increase in pH3 levels in WT MEF cells and MEF -/- cells stably expressing empty vector (EV) or Iso-2, with no change in pH3 levels in MEF-/- cells expressing S6K1. We interpret this as MEF-/- cells expressing S6K1 are stalled in G2 phase and are not entering M phase. Detection of Cyclin A and B levels allows us to distinguish between S/G2 and M phases. We observe no difference in S phase before and after IR in all cells, while we do see an increase in G2 fraction in MEF-/- cells expressing S6K1 after IR (new Figure 5). Panels a and b from Figure 5 in the original manuscript were shifted now to new Figure 5—figure supplement 1.

If there is a G2 checkpoint being activated, is it dependent on ATM, ATR, and/or Wee1 (could use inhibitors to investigate)?

We did not observe any difference in phosphorylated ATM in cells with and without S6K1 (MEF and MCF10A cells) after γ or UV damage (Figure 1—figure supplement 3). We therefore did not investigate this pathway further.

3) Effects of gene expression or knockout on HR rates in Figure 6 can be confounded by effects on S phase fraction. The authors should include analysis of S phase fraction for each of the conditions being evaluated, particularly since their proposed mechanism of action may involve cell cycle regulation.

We agree with the reviewer that effect on HR rates can be confounded by effects on S phase fraction. Similar to point 2 above, we have now repeated the cell cycle experiments to include mitosis-specific marker (p-H3), S phase marker (cyclin A) and G2 phase marker (cyclin B) in U2OS-DR-GFP cells with knockout and overexpression of S6K1. The use of the cyclin markers allows us to distinguish between G2 and M fractions based on cyclin levels as detected by FACS. We have now included cell cycle analysis using U2OS-DR-GFP cells either overexpressing S6K1 or Iso-2; or overexpressing Cdk1 WT or Cdk1 39A; or overexpressing S6K1 with either Cdk1 WT or Cdk1 39A. We did not observe any difference in cell cycle in the cells with either Cdk1 WT or Cdk1 39A (Figure 6—figure supplement 1, see reply to Reviewer #1, comment 7). We therefore can conclude that the effects on HR are not confounded by effects on S phase fraction.

Minor critiques:1. The authors frequently reference amplification of RPS6KB1 in a subset of breast cancers. It would be helpful if the authors can comment on whether this amplification is associated with overexpression of the full length versus short form transcript?

The amplification of RPS6KB1 in a subset of breast cancers amplifies a relatively large region. It is written in page 3 of the manuscript "However, this amplification covers over 4MB and contains nearly 50 genes^14^ including Tbx2, PRKCA, Tlk2, TUBD1 and PPM1D, that are known for their correlation to breast cancer and/or contribution to oncogenic signaling^14–21^, making estimation of the contribution of S6K1 amplification to the disease difficult."

2. The schema in Figure 1a could be modified to better illustrate that the kinase domain of Iso-2 is truncated.

The figure has been modified (see new Figure 1a).

3. Figure 1e x-axis label seems to have typos.

Corrected.

4. The separation of Msh6 functional studies between Figure 4 and Figure 6f-h seems unnecessary – my preference would be to bring those together.

For better flow of the manuscript, the order of the figures has remained the same.

Reviewer #3.Major comments:1. Though the interactions of Iso-2 with CDK1 and MSH6 were well validated, there were no direct validation of the interactions of S6K1 with CDK1 and MSH6; while the authors proceeded to investigate the regulation of CDK1, MSH6 and others by S6K1 but not Iso-2. Therefore, the authors should provide some explanation of this gain the manuscript.

We apologize for the missing description of the rationale. The interaction of S6K1 with CDK1 and MSH6 is difficult to capture due to possible repulsion of S6K1 interactors by the negative charge added following phosphorylation. Whereas, the interaction with Iso-2 is easier to capture due to the lack of kinase activity. This is explained on page 6 of manuscript "As phosphorylation usually causes conformational changes of the substrate due to steric repulsion of the negatively charged phosphate groups, we hypothesized that the S6K1 kinase inactive variant^7^, short isoform Iso-2, may have a stronger interaction with all putative substrates". This is now clarified again in the revised manuscript on page 7 "It should be noted that the interaction with Iso-2 is stronger due to lack of steric repulsion caused by S6K1 phosphorylation".

2. MSH2 was found to interact with both S6K1 and Iso-2 and seems a better candidate to study the roles for S6K1 in MMR. Why MSH6 but not MSH2 for further exploration?

The reviewer is correct in noting that MSH2 was found to interact with both S6K1 and Iso-2. The MSH2 and MSH6 proteins together form a heterodimer, MutSα, that is a mismatch recognition factor. Both are good candidates for further study. However, due to limited nuclear localization of MSH2 and technical difficulties in expressing MSH2 in the in vitro kinase assay (very big protein), its association with S6K1 was not studied further and we focused our study on MSH6.

Are there no S6K1 phosphorylation motifs (RXXS) in MSH2?

MSH2 has 3 S6K1 phosphorylation motifs. As mentioned above, if there were no limitations of availability of good constructs and expression/cloning possibilities of this big protein, it would have been possible to study it as S6K1 substrate.

3. The authors claimed that their observed functions of S6K1 depend on the kinase activity of S6K1, but they did not show the activity of S6K1 after DNA damages (pS6K1-T389 or its direct target pS6-S235/236).

The reviewer is correct in pointing out that we did not show the activity of S6K1 on pS6K1-T389 or pS6S235/236 after DNA damage. We have now performed an experiment to look at this activity. We have performed a time course experiment on MEF S6K1-/- cells transduced with either pB(-), S6K1 or Iso-2. The cells were irradiated with UV and harvested at different time points. We observe a transient increase in phosphorylation of S6K1 T389 5 minutes after UV damage and an increase in phosphorylation of S6 S235/S236 10 minutes to one hour after UV in cells overexpressing S6K1. We did not observe this phosphorylation of S6 in cells transduced with either pB(-) or Iso-2 (The cells are S6K1/2^-/-^ so there is no endogenous phosphorylation of S6) (Author response image 3).

**Author response image 3. sa2fig3:** Time course of phosphorylation of S6K1 and S6 after UV irradiation. (a) MEK S6K-/- cells transduced with either bB(-), S6k! or Iso-2 were irradiated with UV 30J/m2/sec. Cells were harvested at different time points after irradiation. Western Blot analysis shows phosphorylated S6K1 T389 (pS6K1), total S6K1 (tS6K1), phosphorylated S6 S235/236 (pS6) and total S6 (tS6) levels. Tubulin was used as a loading control. Graph shows relative phosphorylation od S6 and S6K1 in MEF S6K1-/- cells overexpressing S6K1.

Moreover, does DNA damage agents increase the binding of S6K1 to CDK1, MSH6 and others as identified by mass spectrometry analyses?

This experiment was not performed. The mass spectrometry analysis was only performed on untreated cells.

Does DNA damage agents increase the phosphorylation of CDK1 and MSH6 by S6K1?

DNA damaging agents activate or inactivate the mTOR pathway, depending on the agent and cell type. There is strong evidence for the effects of mTOR inhibition on the DNA damage response and little about the role of S6K1. Moreover, the direct roles of mTOR or S6K1 in regulating the DNA repair machinery are mostly unknown. We show here that cells that are knocked out for S6K1 are more sensitive to DNA damaging agents because of lower DNA repair and vice versa for S6K1 overexpression. Measuring Cdk1 and MSH6 phosphorylation in whole cell lysates is impossible due to lack of specific phospho-antibodies (the one commercial phospho-Cdk1 antibody is not specific, we tried it). We now present an immunoprecipitation experiment with 293 cells transfected with either empty vector (-), S6K1 or co-transfected with S6K1 and Cdk1, before and after UV irradiation. We observed an accumulation of phosphorylation on Cdk1 (as seen by IP of HA-Cdk1 and Western blot with antibody against pAKT substrate) in cells co-transfected with Cdk1 and S6K1 after UV irradiation (new Figure 3e and f). Unfortunately, we were unable to perform similar experiments with MSH6 due to technical issues.

4. The authors claimed that S6K1 does not affect ATM levels or phosphorylation state, which were mainly based on the results as shown in Supplementary Figure 3. However, most of the results in Supplementary Figure 3 are not convincing: (1) The blots of loading control GAPDH were very poor indicating not equal protein loading during immunoblotting; (2) In Supplementary Figure 3e and 3g, it seems that there was relative increase of ATM in S6K-/- MEFs compared to wild type MEFs. The authors need to provide better immunoblots.

We have now quantitated the Western blots from three biological replicates and present this analysis in Figure 1—figure supplement 3a-h.

Minor comments:1. Figure 1a. please indicate which isoform of S6K1 (p85S6K1 or p70S6K1) was used for Halo tagged proteins. It seems to be p85S6K1.

The reviewer is correct, it is now made clear in Figure 1a.

2. Figure 4c. there were no apparent difference of nuclear MSH2 among mlp(-), sh1 and sh2 groups after UV, but the authors said '….. cells with S6K1 knockdown showed reduced nuclear localization of MSH2 and MSH6 following UV irradiation…'(page 8) in the text. Please clarify.

The text has now been corrected to " … cells with S6K1 knockdown showed reduced nuclear localization of MSH6 following UV irradiation…"

3. There was no text description of Supplementary Figure 7a,b in the Results part.

We apologize for this omission. This has now been included. "We observed increased GFP levels in the S6K1 expressing U2OS-DR-GFP cells as compared to control cells or Iso-2 expressing cells (Figure 6b) with no change in cell cycle (Figure 6—figure supplement 1a-d)".

4. Page 14 line 12: please change '…PKC36.' to '…PKC^36^.'

We thank the reviewer – Corrected.

5. Page 14 line 20: please delete one 'that' in the sentence '…rule out that the phosphorylation…'.

We thank the reviewer – Corrected.